METHODS AND RESOURCES

# Targeting AAV vectors to the central nervous system by engineering capsid–receptor interactions that enable crossing of the blood–brain barrier

Qin Huang[1], Albert T. Chen[1,2], Ken Y. Chan[1], Hikari Sorensen[1], Andrew J. Barry[1], Bahar Azari[3], Qingxia Zheng[1], Thomas Beddow[1], Binhui Zhao[1], Isabelle G. Tobey[1], Cynthia Moncada-Reid[1], Fatma-Elzahraa Eid[1,4], Christopher J. Walkey[5], M. Cecilia Ljungberg[6,7], William R. Lagor[5], Jason D. Heaney[8], Yujia A. Chan[1], Benjamin E. Deverman [1]*

1 Stanley Center for Psychiatric Research, Broad Institute of MIT and Harvard, Cambridge, Massachusetts, United States of America, 2 Biological and Biomedical Sciences, Harvard University, Cambridge, Massachusetts, United States of America, 3 Electrical & Computer Engineering Department, Northeastern University, Boston, Massachusetts, United States of America, 4 Department of Systems and Computer Engineering, Al-Azhar University, Cairo, Egypt, 5 Department of Integrative Physiology, Baylor College of Medicine, Houston, Texas, United States of America, 6 Department of Pediatrics, Baylor College of Medicine, Houston, Texas, United States of America, 7 Duncan Neurological Research Institute at Texas Children's Hospital, Houston, Texas, United States of America, 8 Department of Molecular and Human Genetics, Baylor College of Medicine, Houston, Texas, United States of America

* bdeverma@broadinstitute.org

**Data Availability Statement:** All code used in this study is available on GitHub: https://github.com/vector-engineering/AAV_capsid_receptor/. All data

## Abstract

Viruses have evolved the ability to bind and enter cells through interactions with a wide variety of cell macromolecules. We engineered peptide-modified adeno-associated virus (AAV) capsids that transduce the brain through the introduction of de novo interactions with 2 proteins expressed on the mouse blood–brain barrier (BBB), LY6A or LY6C1. The in vivo tropisms of these capsids are predictable as they are dependent on the cell- and strain-specific expression of their target protein. This approach generated hundreds of capsids with dramatically enhanced central nervous system (CNS) tropisms within a single round of screening in vitro and secondary validation in vivo thereby reducing the use of animals in comparison to conventional multi-round in vivo selections. The reproducible and quantitative data derived via this method enabled both saturation mutagenesis and machine learning (ML)-guided exploration of the capsid sequence space. Notably, during our validation process, we determined that nearly all published AAV capsids that were selected for their ability to cross the BBB in mice leverage either the LY6A or LY6C1 protein, which are not present in primates. This work demonstrates that AAV capsids can be directly targeted to specific proteins to generate potent gene delivery vectors with known mechanisms of action and predictable tropisms.

required to reproduce each plot are available from the Zenodo open repository with DOI: 10.5281/zenodo.7689795. These data include RPMs and enrichment values from the libraries used in this study. Supplementary Data contain analyzed and processed values derived from the data available on Zenodo. Some of the data presented in S9 Fig is available as part of the NIH SCGE Toolkit at https://scge.mcw.edu/toolkit/data/experiments/group/1441 All materials were acquired from commercial vendors as described. Packaging plasmids carrying the individually characterized LY6A (AAV-BI28: 203532; AAV-BI48: 203533; AAV-BI49:203534) or LY6C1 binding capsids (AAV-BI28: 203532; AAV-BI62:203535; AAV-BI65:203536), AAV-CAG-NLS-mScarlet-2A-Luc-WPRE-pA (203539), AAV-GfABC1D-SaCas9-WPRE-pA (203540), and AAV-GfABC1D-GFP-U6-L1-U6-R2 (203541) will be made available through Addgene under the indicted Addgene IDs.

**Funding:** Work in this study was supported by the National Institute of Neurological Disorders and Stroke and NIH Common Fund through the Somatic Cell Genome Engineering (SCGE) Program (UG3NS111689 to B.E.D); the Stanley Center for Psychiatric Research, Broad Institute (B.E.D); Apertura Gene Therapy (B.E.D) a Brain Initiative award funded through the National Institute of Mental Health (UG3MH120096 to B.E.D). F.E.E. was supported by a Broad Shark Tank Award and Y.A.C. was supported by a Broad Ignite Award. Work at the BCM-Rice was supported by NIH SCGE grant (U42OD026645 to J.D.H. and W.R.L). The funders had no role in study design, data collection and analysis, decision to publish, or preparation of the manuscript.

**Competing interests:** I have read the journal's policy and the authors of this manuscript have the following competing interests: BED is a scientific founder at Apertura Gene Therapy and a scientific advisory board member at Tevard Biosciences. BED, QH, KYC, and FEE are named inventors on patent applications filed by the Broad Institute of MIT and Harvard related to this study. Remaining authors declare that they have no competing interests.

**Abbreviations:** AA, amino acid; AAV, adeno-associated virus; BBB, blood–brain barrier; CNS, central nervous system; ML, machine learning; NGS, next-generation sequencing; OCT, optimal cutting temperature; PBS, phosphate buffered saline; PWM, position weight matrix; RPM, reads per million; SVAE, supervised variational auto-encoder.

## Introduction

Gene therapy with recombinant adeno-associated viruses (AAVs) shows promise for treating diseases at their root genetic cause, but remains constrained by the inefficiency of delivery to disease-relevant organs and cell types. Natural AAV capsids can be modified to produce vectors with dramatically improved in vivo tropisms. An effective engineering strategy has been to generate diverse libraries of capsid variants via peptide insertions and to subject these libraries to multiple rounds of in vivo selection to identify capsids with the desired properties such as central nervous system (CNS)-wide transduction [1–3], brain vascular endothelium targeting [4,5], retrograde transduction in the CNS [6], transduction of human hepatocytes in a xenograft system [7], photoreceptor transduction [8], and muscle transduction [9,10]. However, these screening efforts have been limited to function-focused approaches, where capsids are selected for a particular biodistribution or cell type tropism without discriminating for mechanism of action. The mechanism underlying the selected function must typically be elucidated via detailed downstream studies. As a result, the high-performance capsids identified in these extensive screens in animals may often rely on mechanisms of action that are not conserved across species [5,7,11–13]. Human cell or organoid models of increasing sophistication may provide new opportunities for human-relevant capsid engineering [14–19]. However, without a clear and preserved underlying mechanism of action, capsids selected in vitro may not retain their selected function in vivo.

Several research groups have attempted to circumvent this shortcoming by innovating mechanism-focused approaches, e.g., grafting independently characterized or engineered peptides [20–22] or proteins, such as DARPins or antibody fragments, onto the AAV capsid [23–29]. However, these grafting approaches do not select for optimal affinity in the context of the functional vector and may increase the complexity of manufacturing. Therefore, the majority of AAV capsid engineering efforts to date have continued to focus on in vivo selections.

In 2019, we and others reported that AAV-PHP.B [2] and related capsids [1–3] could utilize a novel blood–brain barrier (BBB)-crossing mechanism by interacting with the LY6A protein on the surface of the brain endothelium of a subset of mouse strains [11,12]. Based on this finding, we were encouraged to develop a novel mechanism-focused approach that screens an AAV capsid library for variants that bind cellular proteins that are likely to translate into a desired in vivo tropism—in this case, BBB-crossing activity. As a proof-of-concept, we targeted 2 mouse CNS endothelium proteins, LY6A and LY6C1 [11,12], and used pull-down assays to screen for AAVs capable of directly binding these target proteins in vitro. A large fraction of the capsids engineered to interact with LY6A or LY6C1 in vitro exhibited in vivo BBB-crossing activity that was enhanced relative to AAV9 and comparable to other reported capsids with improved CNS tropisms. In addition, in our validation process, we found that nearly all published AAV capsids (24 out of 26 tested) that were selected for BBB-crossing in mice [1–3,30,31] rely on either the LY6A or LY6C1 protein. As these proteins do not have known homologs in primates, it is unlikely that the enhanced CNS tropisms of LY6A or LY6C1-targeting capsids will translate from mice to primates. A key advantage exhibited in our approach was that it generated highly quantitative and reproducible data from a single round of screening in vitro; this enabled rapid motif identification and the generation of a diverse set of additional sequences, through saturation mutagenesis and machine learning (ML), that were found to exhibit high levels of CNS transduction in vivo. In contrast to the large body of AAV engineering studies that have leveraged in vivo selections to identify AAV capsids with enhanced tropisms but unknown mechanisms of action and unpredictable translatability across species, this work demonstrates that capsids can be systematically targeted to defined cell surface proteins to facilitate enhanced and predictable in vivo tropisms.

## Results

### A high-throughput purified protein assay identifies capsids selective for LY6A or LY6C1

To assess the potential of a mechanism-focused approach to develop capsids with enhanced CNS tropisms, we targeted 2 surface proteins present on brain vascular endothelial cells: LY6A, the known receptor for the AAV-PHP.B family of capsids, as a positive control, and a related protein, LY6C1, a novel target likewise highly expressed on CNS endothelial cells [11,32]. LY6C1 was selected based on the hypothesis that it may share LY6A's ability to mediate AAV transport into the CNS, given that the LY6 family possesses a conserved protein structure and subcellular localization [33]. We generated LY6A and LY6C1 proteins as Fc fusions and used a magnetic bead-based pull-down assay to perform initial (Round 1) screens of 2 independently generated 7-mer-modified AAV9 libraries (random 7-mer amino acid sequences were inserted between residues 588–589 in VP1)—named Library 1 and Library 2, respectively—for variants that bind to LY6A-Fc, LY6C1-Fc, or an Fc-only control (Library 1 data is shown in Fig 1B, 1D and 1E; Library 2 data is shown in S1 Fig). In the libraries, each capsid variant packaged its own capsid-encoding genome, allowing for the assessment of binding to the target by short-read, next-generation sequencing (NGS). The pull-down assays yielded reproducible binding scores for capsid variants across a wide dynamic range, with a high correlation of read depth-normalized counts between replicates (Figs 1B and S2A–S2H).

To compare the pull-down assays to a conventional in vivo selection, we screened Library 2 for capsids that transduced the C57BL/6J and BALB/cJ CNS ($n$ = 2 mice per strain) using transcribed capsid sequences as a functional readout [3,4]. The vast majority of variants were detected in only 1 animal (Figs 1C and S2I and S2J) as has been observed in other in vivo selection experiments [3,9]. In contrast, the pull-down assays yielded thousands of unique capsids that selectively bound the intended target but not the Fc-only control (Figs 1D and S1A) or the other target, i.e., capsids selected for LY6C1 binding were not highly enriched for LY6A binding, and vice versa (Fig 1E).

### A wide array of sequence motifs is enriched by the pull-down assays

To assess the diversity of sequences enriched among LY6A- and LY6C1-binding 7-mers, we projected the protein-specific sequences highlighted in Fig 1D using UMAP [34] and jointly clustered sequences from Library 1 and Library 2 with a Gaussian mixture model (k = 40) (Figs 1F and 1G and S3A and S3B, S1–S6 Data). All clusters for both LY6A and LY6C1 have representatives from both libraries (S3C and S3D Fig). This consistency across independent libraries demonstrated that the approach can reproducibly detect thousands of unique capsid sequences with common sequence motifs. Inspection of the clusters of LY6A- and LY6C1-binding 7-mers revealed clear sequence motifs, typically 2 to 4 amino acids in length. Of these motifs, some were similar to previously published capsid sequences with CNS tropisms (Fig 1F–1I, all clusters are shown in S4 Fig and S1–S6 Data). For example, clusters containing sequences similar to those of known LY6A-binding capsids were observed: AAV-PHP.B (TLAVPFK), clusters A5, 13, 33; AAV-PHP.B2 (SVSKPFL), clusters A14, 18, 32; and AAV-PHP.B3 (FTLTTPK), clusters A17, 35 (S7 Data).

### In vitro selection for LY6A and LY6C1 binding yields capsids with enhanced tropisms predicted by target expression

To test whether the capsids selected to target LY6A or LY6C1 in vitro enable efficient BBB crossing, we generated a Round 2 library containing top hits from the initial Round 1 screen

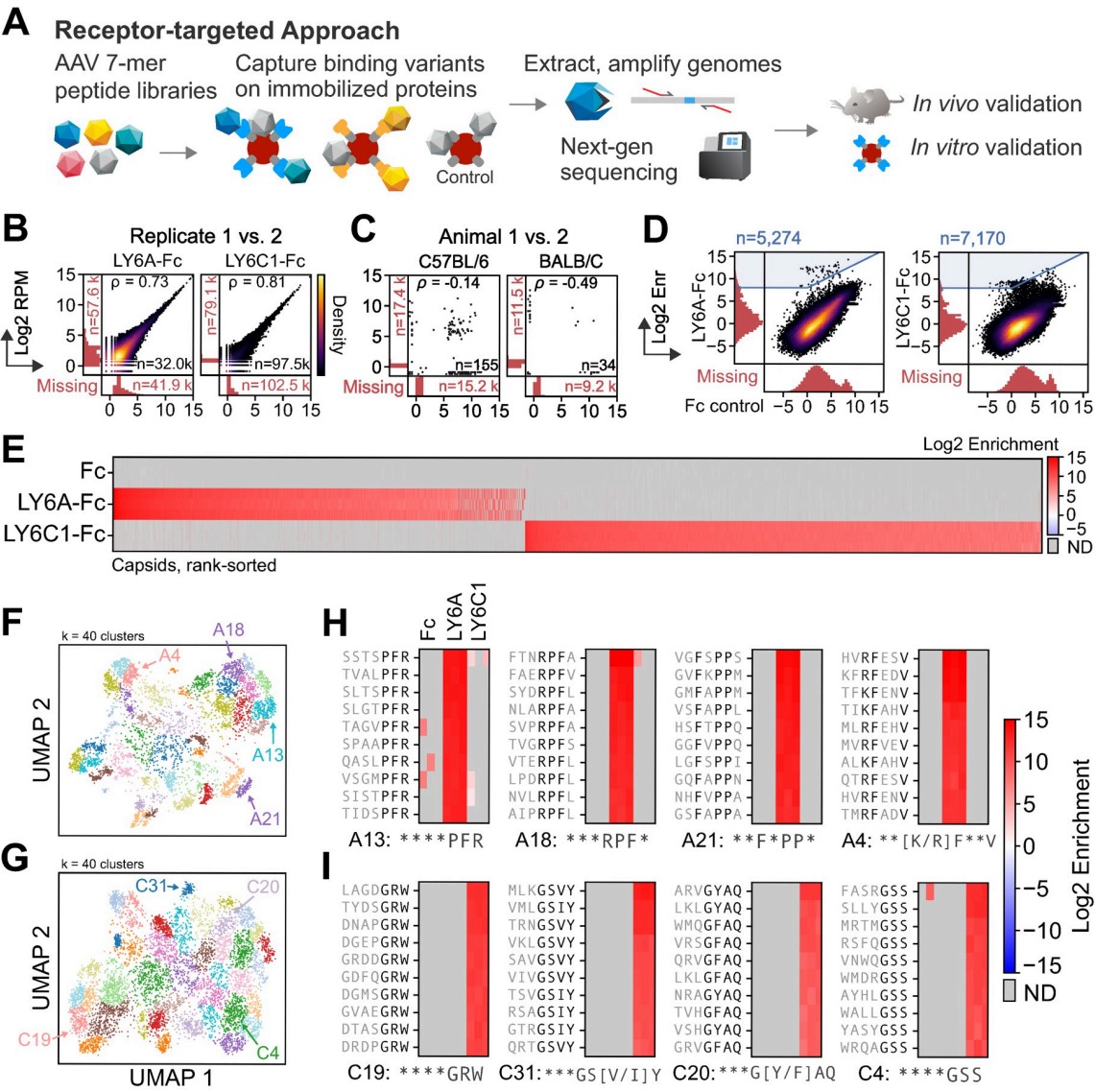

**Fig 1. In vitro pull-down assays yield capsids that selectively bind LY6A or LY6C1.** (A) A capsid library is screened for the ability to bind immobilized target Fc-fusion proteins. Bound capsid sequences are extracted and subjected to NGS. Hits are incorporated into a focused library for in vivo and in vitro validation. (B, C) The Pearson correlation of the $\log_2$ normalized read count (RPM) are shown between biological replicates ($n = 3$, only 1 pair shown) (B) and between animals ($n = 2$) (C). Variants detected in 1 replicate or animal and not the other are shown in the marginal histograms. (D) The variant $\log_2$ enrichment (average RPM between replicates, normalized to the starting library RPM) plotted between the target (y-axis) and Fc-only control (x-axis) shows a majority of variants with nonspecific binding and a minority (blue highlighted region) with target-specific binding. The variants detected in 1 assay and not the other are shown in the marginal histograms. (E) The $\log_2$ enrichment of the selected variants highlighted in blue in (D) with each replicate's enrichment plotted in separate rows ($n = 3$). ND = not detected. (F, G) The sequences that bound LY6A (F) or LY6C1 (G) from Libraries 1 and 2 were one-hot encoded, jointly projected with UMAP, and jointly clustered with a Gaussian mixture model ($k = 40$, S1–S6 Data). (H, I) Four clusters for each target from (F, G) were manually selected based on whether there was a clear motif 2–4 amino acids in length that matched either existing reference sequence (LY6A-binding: ***PFR, ***RPF, LY6C1-binding: ***G[Y/F]AQ) or represented a motif not yet seen in published studies. Consensus motifs are defined per-position, with flexible amino acid residues (asterisks) and fixed residues (present in more than 40% of the cluster's sequences; black letters). The underlying data supporting Fig 1B and 1D and 1E can be found at https://doi.org/10.5281/zenodo.7689794: library1.csv; Fig 1C at https://doi.org/10.5281/zenodo.7689794: library2_invivo.csv; Fig 1F and 1H at https://doi.org/10.5281/zenodo.7689794: LY6A_joint_umap_l1_l2.csv; Fig 1G and 1I at https://doi.org/10.5281/zenodo.7689794: LY6C1_joint_umap_l1_l2.csv. NGS, next-generation sequencing; RPM, reads per million.

for LY6A and LY6C1 binding ($n$ = 6.4K and 12.6K unique 7-mers, respectively). To compare the pull-down assay approach to conventional in vivo selections, we included all unique 7-mer sequences recovered following the Round 1 screening for expression in the CNS of C57BL/6J or BALB/cJ mice ($n$ = 5.8K) (Fig 2A; only Library 2 was used in the Round 1 in vivo screens) as well as a panel of published reference capsids from prior in vivo selections for CNS transduction [1–3,30,31]. The references included members of the AAV-PHP.B family known to utilize the LY6A receptor to cross the BBB [11,35] (S8 Data). Each 7-mer amino acid (AA) sequence in the library was encoded by 2 nucleotide sequences, which served as biological replicates.

The Round 2 library was screened in vitro and in vivo as in Round 1 (Fig 2A). The Round 2 data showed high agreement between replicates both in vitro and in vivo (S5 Fig) and between 7-mer AA replicates across all assays (S6 Fig). The majority of sequences identified in the Round 1 pull-down assays were validated by selective binding to their expected target in the Round 2 library screens (Fig 2B). When assessed for their ability to transduce cells in the CNS of C57BL/6J or BALB/cJ mice, hundreds of the capsid sequences identified by the pull-down assays were highly enriched ($\log_2$ enrichment > 4; Fig 2B and 2C). In comparison, far fewer sequences identified in the Round 1 in vivo screen were enriched in the Round 2 screen (Fig 2B and 2C). As previously observed for the AAV-PHP.B family, LY6A-binding capsids were highly enriched for in vivo transduction in the brains of C57BL/6J mice, but not BALB/cJ mice (Fig 2C and 2D). In contrast, numerous LY6C1-binding capsids were highly enriched in both mouse strains. These findings align with the levels of the target protein in each mouse strain [11].

We ranked the capsids identified by the pull-down assays and in vivo screening in Round 1 based on their enrichment in the CNS of C57BL/6J or BALB/cJ mice in Round 2 (Fig 2D). The ranking included previously characterized reference capsids such as AAV-PHP.B and AAV-F. Numerous capsids identified through the pull-down assays ranked among the top performing capsids in the in vivo selection alongside the reference capsids (Fig 2E). Notably, of 26 reference capsids identified in 4 prior independent studies using 3 different selection strategies in mice, 24 bound to LY6A or LY6C1 in vitro; 9P31 and 9P36 [3] did not detectably bind to either LY6A or LY6C1 under our assay conditions (Fig 2F). Newly published mouse BBB-crossing AAV capsids, MDV1A, MDV1B, and M.Mus.1, which did not exhibit enhanced CNS tropisms in macaque, also possess motifs observed among the LY6C1-binding variants in our collection [13]. These results suggest that LY6A and LY6C1 are capable of efficiently mediating the transportation of AAVs into the mouse CNS and that engineering capsids to bind proteins with such abilities can be an effective strategy to enhance tropism in a predictable manner based on target expression.

## Identification of a cluster of brain-enriched capsids from the Round 2 in vivo screen

The best performing CNS-transducing capsids in our Round 2 in vivo screen were LY6A or LY6C1 binders; however, we investigated the small subset of CNS-transducing capsids from the top Round 1 in vivo hits that did not bind to either target in the Round 2 pull-down assays (S7A Fig, S9 Data, $\log_2$ enrichment LY6A-Fc < 0, LY6C1-Fc < -2, C57BL/6J or BALB/cJ > 2, $n$ = 180; these capsids did not pass the more stringent in vivo enrichment cutoff implemented in Fig 2C). Clustering of these capsids by pairwise hamming distance resulted in many small clusters and 1 larger cluster (S7B Fig). The large cluster had generally high BALB/cJ CNS transduction but lower efficiency in C57BL/6J and exhibited a clear motif of *N*[T/V/I][R/K]** (S7C and S7D Fig). Sequences in this cluster resemble that of our recently published

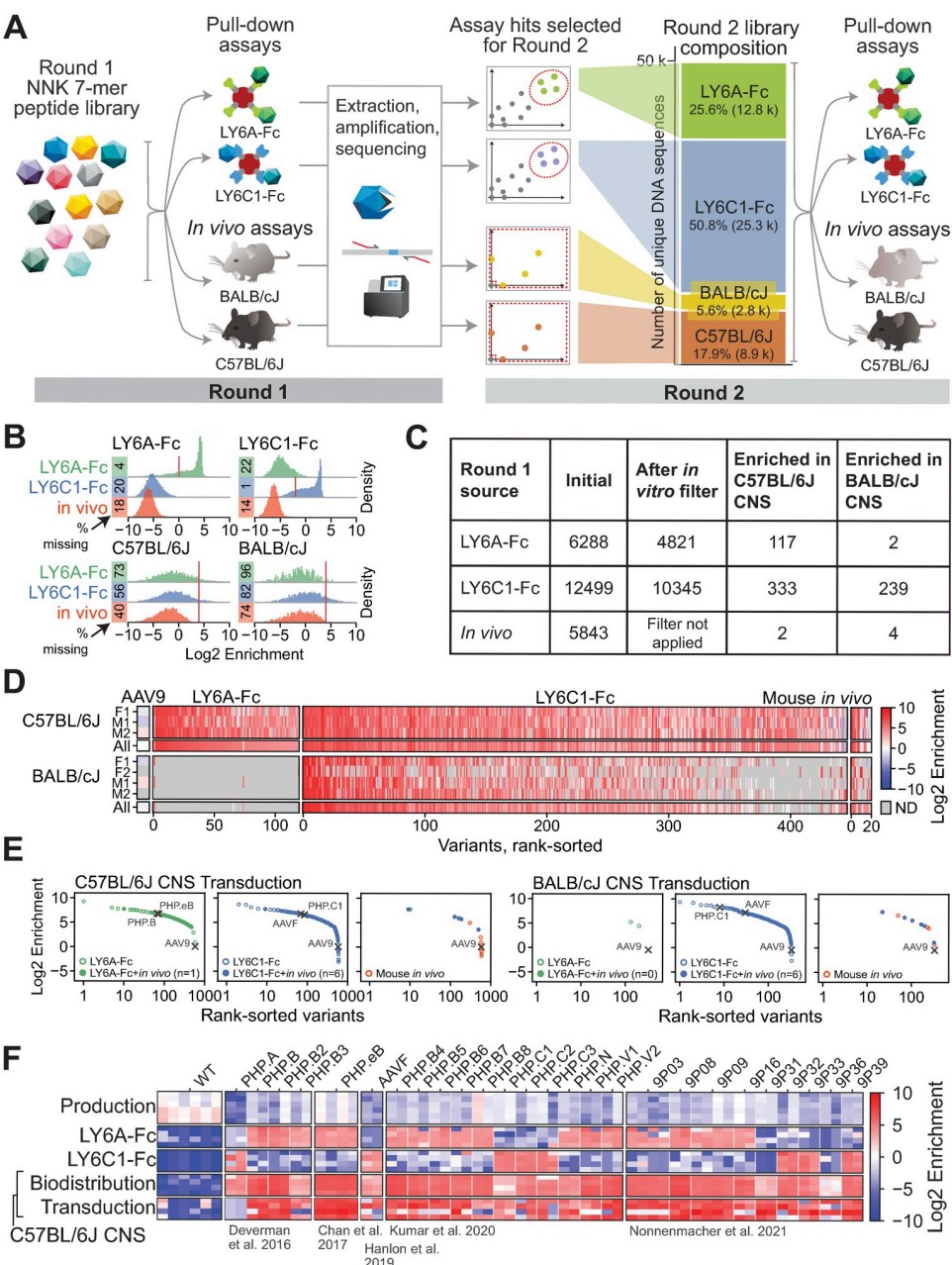

**Fig 2. Round 2 validation of LY6A- and LY6C1-binding variants identified thousands of capsids with CNS transduction activity.** (**A**) The Round 2 library was composed of a selection of top performing variants from the Round 1 assays for LY6A binding, LY6C1 binding, in vivo CNS transduction, and published reference sequences. The Round 2 library was subjected to screening as in Round 1. (**B**) The distributions show the Round 2 library performance in the pull-down assays and the CNS transduction of BALB/cJ and C57BL/6J mice, grouped and colored by the Round 1 selection source of each variant. The red lines indicate the thresholds set for the filters applied in (**C**). (**C**) The hits in Round 2 were identified as follows: target-binding capsids from the Round 1 screen were first filtered on their respective target binding activity in the Round 2 screen (LY6A: log$_2$ enrichment > 0, LY6C1: log$_2$ enrichment > -2). Variants were then filtered on Round 2 in vivo CNS transduction (log$_2$ enrichment > 4 in either mouse strain and detected in at least 2 animals within that strain). (**D**) The in vivo log$_2$ enrichment scores in C57BL/6J and BALB/cJ mice of Round 2 library variants that were filtered for high in vivo log$_2$ enrichment scores in (**C**). The scores in individual animals (M*, F*) for each strain are shown alongside the average across animals (all). Variants are shown grouped and colored by their Round 1 selection source and rank-sorted on a combined score of C57BL/6J and BALB/cJ transduction. (**E**) The filtered variants from (**C**) are shown grouped and colored by their Round 1 selection source and rank-sorted separately for each mouse strain. Reference controls and AAV9 are marked with crosses. Variants

identified in both the Round 1 pull-down assays and in vivo screen are displayed as filled dots. (**F**) The target binding and C57BL/6J CNS biodistribution or transduction phenotypes of reference capsids with CNS tropisms are shown. Each capsid is represented by at least two 7-mer AA replicates (each column indicates a separate replicate). The underlying data supporting Fig 2A can be found at https://doi.org/10.5281/zenodo.7689794: round2_codons_separate. csv; Fig 2B–E at https://doi.org/10.5281/zenodo.7689794: round2_codons_merged.csv; Fig 2F at https://doi.org/10. 5281/zenodo.7689794: SVAE_SM_library_references_only.csv. AA, amino acid; AAV, adeno-associated virus; CNS, central nervous system.

AAV-BI30 capsid (AAV9 with the 7-mer insertion NNSTRGG), which highly transduces endothelial cells throughout the CNS of multiple mouse strains and rats in vivo, as well as human brain microvascular endothelial cells in vitro [4].

## AAV capsids developed via pull-down assays effectively deliver genes to the mouse CNS

Top hits from the Round 2 in vivo selection were nominated for individual in vivo testing in BALB/cJ and C57BL/6J mice. First, we clustered the variants in the LY6A- and LY6C1-binding subsets from the Round 2 library that exhibited a $\log_2$ enrichment of more than 0 or −2, respectively (Fig 3A, S10 Data). Not all LY6A-binding clusters yielded capsids that were enriched in the C57BL/6J brain in the Round 2 in vivo selection (Fig 3B). To test sequences in different clusters identified via the pull-down assays, 5 variants were selected for individual characterization based on their (1) mean brain transduction enrichment scores; (2) consistency of observed enrichment across replicates in the Round 2 screens (Figs 3C and S8A and S8B); (3) sequence diversity (the variants AAV-BI48, AAV-BI49, AAV-BI28, AAV-BI62, AAV-BI65 each represent different clusters as shown in Fig 3A); and (4) production fitness (estimated from the enrichment of the variants in the virus library compared to the plasmid library). None of the capsids chosen for individual characterization were more detargeted from the liver than previously published variants with CNS tropisms (S8C Fig). When individually administered in mice, each variant exhibited enhanced CNS transduction compared to AAV9 that was consistent with their mechanism of action; AAV-PHP.eB, AAV-BI48 and AAV-BI49, which bind to LY6A, exhibited an enhanced CNS tropism in only C57BL/6J mice, whereas AAVF, AAV-BI28, AAV-BI62, and AAV-BI65, which bind to LY6C1, maintained their enhanced tropism across both mouse strains (Fig 3D).

Since previously published LY6A-binding variants such as AAV-PHP.B and AAV-PHP.eB have been carefully characterized [1,2,11], we chose a representative LY6C1-binding variant, AAV-BI28, for further study. When administered to adult C57BL/6J mice, we observed transduction of NeuN+ neurons, S100+ astrocytes, CC1+ oligodendrocytes (S9A–S9C Fig). We next evaluated the use of AAV-BI28 for gene editing in astrocytes throughout the Ai9 (Cre recombinase-dependent tdTomato reporter) mouse brain [36]. We built a two-vector system with the first vector expressing SaCas9 under the control of a truncated GFAP promoter (AAV-BI28-GfABC₁D-SaCas9) and the second vector expressing 2 gRNAs and green fluorescent protein (AAV-BI28:GfABC₁D-NLS-GFP-2x-U6-gRNA) (S9D Fig). The 2 gRNAs were designed to target SaCas9 to sequences flanking the stop cassette that prevents expression of the tdTomato reporter. The 2 vectors were administered intravenously into adult Ai9 mice and gene editing at the reporter locus (tdTomato) was assessed in cortical, thalamic, and striatal astrocytes 4 weeks later. Our results demonstrate that BI28 can be used to induce gene editing in a substantial fraction of astrocytes in the adult mouse brain (S9E–S9H Fig). These data and supporting in vitro data are available as part of the NIH Somatic Cell Genome Editing (SCGE) Program Toolkit (https://scge.mcw.edu/toolkit/data/experiments/group/1441) and

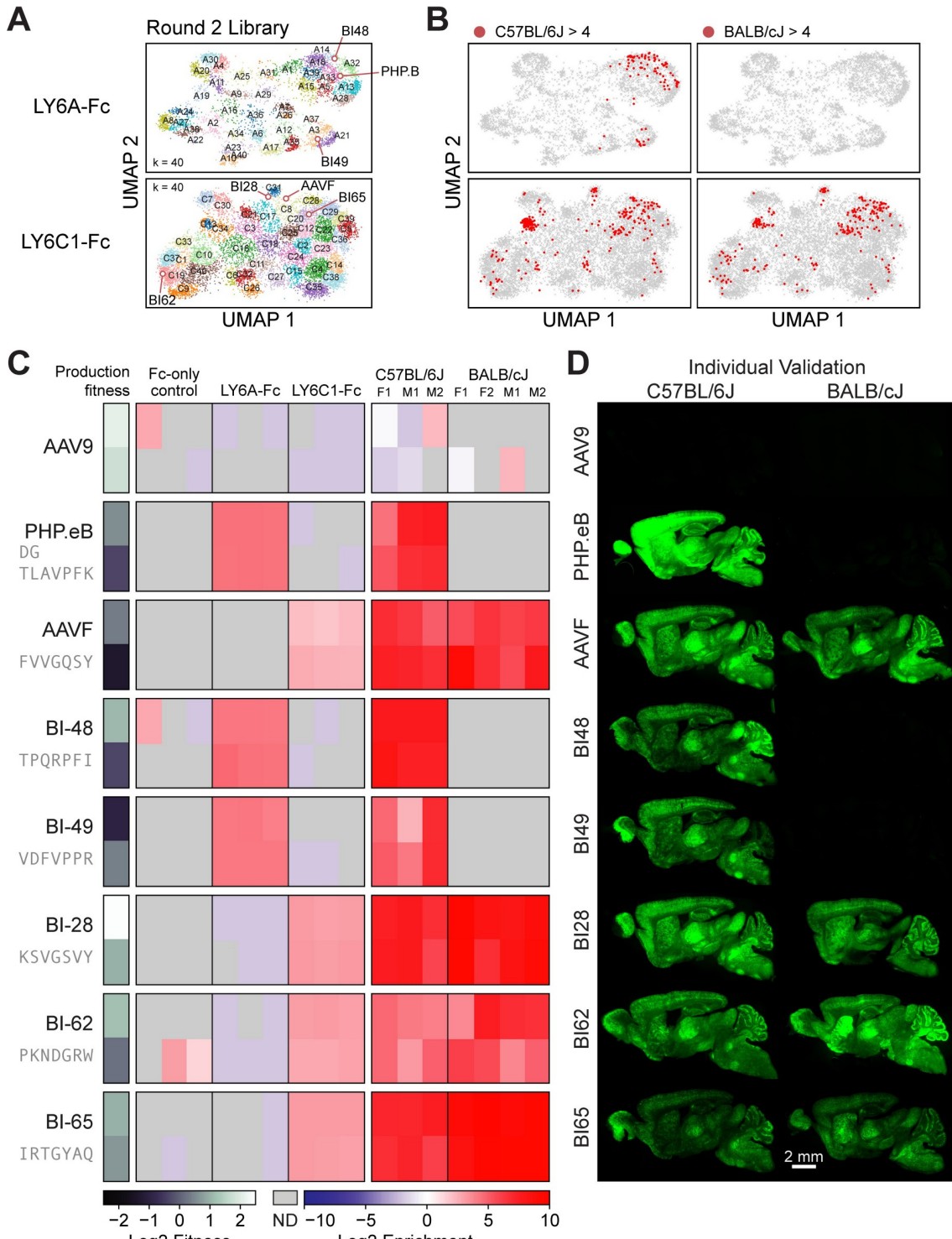

**Fig 3. LY6A- and LY6C1-binding capsids identified in the pull-down assays cross the mouse BBB.** (A) The UMAPs of Round 2 library variants are shown projected onto the UMAPs of Round 1 variants. Variant sequences were clustered with K-means (LY6A, k = 25; LY6C1, k = 30) (see cluster summaries in S10 Data). (B) The Round 2 variants with an in vivo brain transduction log2 enrichment of > 4 in C57BL/6J mice (left) and BALB/cJ mice (right) are marked in red. (C) The Round 2 in vivo screen results for the reference capsids and 5 Round 2 variants selected for individual characterization are shown. Each variant is represented by two 7-mer AA replicates indicated by separate rows. ND = not detected. (D) Representative brain images are shown for the capsids in (C) that were individually tested in C57BL/6J mice (left) and BALB/cJ mice (right). The underlying data supporting Fig 3A and 3B can be

found at https://doi.org/10.5281/zenodo.7689794: round2_codons_merged.csv; Fig 3C at https://doi.org/10.5281/zenodo.7689794: round2_codons_separate.csv. AA, amino acid; BBB, blood–brain barrier.

were reproduced independently by the Baylor College of Medicine-Rice Small Animal Testing Center (BCM-Rice SATC) organized as part of the NIH SCGE Consortium (S9D–S9H Fig).

## The pull-down assay approach yields replicable, quantitative data that enables machine learning-guided sequence diversification

Screens for capsids with functions of interest typically only sample a small fraction of the theoretical sequence space (for a 7-mer insertion, the amino acid sequence space is $20^7$ or 1.28 billion). While it is impractical or even impossible (especially for longer sequences) to experimentally assay substantial portions of the sequence space, it is possible to train ML models using limited assay data to extend predictions to the rest of the unassayed sequence space. The highly replicable and quantitative pull-down assay data are amenable to ML-guided approaches for mapping 7-mer sequences to target binding.

To generate more diverse target-binding sequences, we sought to evaluate an ML-guided approach to generate more diverse target-binding sequences training on data from only a single round of screening. We designed a library containing novel sequences generated using a supervised variational auto-encoder (SVAE) ML model or by saturation mutagenesis around specific motifs (Figs 4A and S10A). As the Round 2 library was produced and assayed separately from the SVAE and saturation mutagenesis library, we used control sequences included in both libraries to perform calibration to account for the relative nature of enrichment and for batch effects (S11 Fig, S12–S23 Data). To generate variants via saturation mutagenesis, we chose to explore 1 highly enriched motif identified through LY6A and LY6C1 binding from the Round 1 screen: ***[K/R]PF[I/L] and ***G[W/Y]S[A/S], respectively (32K per motif; Fig 4A). These motifs were chosen as they were formed around residues with similar biochemical characteristics and contained many highly performant variants. The library containing the SVAE- and saturation mutagenesis-generated variants was subjected to pull-down assays and in vivo assays, and its results were compared to those of the Round 2 library.

To generate variants via ML, we used a sequence generation method based on latent-representation-learning models (Figs 4B and S12 and S13), which have previously been applied toward the generation of a diverse set of viable capsids [37]. SVAE models for LY6A and LY6C1 binding were trained using one-hot encodings of 7-mer amino acid sequences and their associated target binding $\log_2$ enrichment. SVAE model accuracy was assessed by predicting binding enrichment on a held-out test set (Pearson correlation $\sigma_A \approx 0.83$ and $\sigma_C \approx 0.85$ for LY6A and LY6C1, respectively). Novel sequences with high predicted target binding were generated by clustering the high enrichment portion of the SVAE latent space, and then sampling from each cluster's position weight matrix (amino acid frequencies of sequences in each cluster) (Figs 4B and S12–S14).

SVAE-generated variants were then evaluated in silico for predicted production fitness to preempt a high proportion of the variants failing to be produced at detectable levels in the virus library [38]. For the saturation mutagenesis approach, we generated all possible 7-mer sequences containing these motifs without filtering by predicted production fitness. The SVAE-generated variants were predicted to be prone to low production fitness (S10B Fig), likely as a result of optimizing solely for binding. Therefore, we generated 2 sets of variants via the SVAE: (1) 4K variants with highest predicted target binding (ignoring production fitness); and (2) 9K variants scoring the highest according to a joint score (Materials and methods, Eq

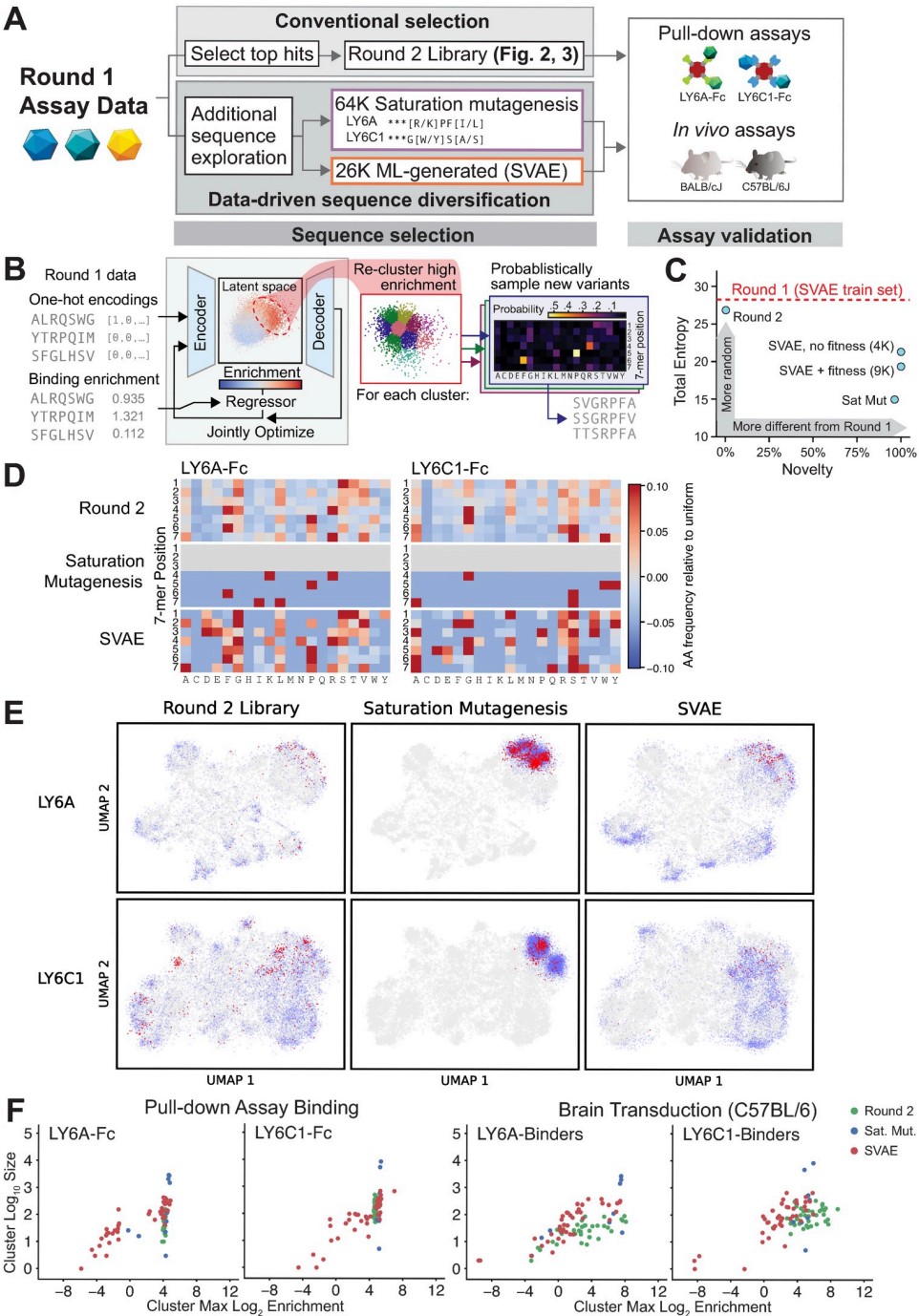

**Fig 4. A single round of screening data can be used with SVAE and saturation mutagenesis to generate additional functional sequences.** (**A**) Round 1 data were used to explore additional sequence diversity via 2 methods: saturation mutagenesis around 2 motifs (LY6A \*\*\*[K/R]PF[I/L], LY6C1 \*\*\*G[W/Y]S[A/S]) and SVAE ML generation. (B) The SVAE was trained on Round 1, library 1 sequences (encoder/decoder blocks) and binding enrichments (regression block). During training, these blocks were jointly optimized. High-binding enrichment sequences were isolated and re-clustered, and new sequences were sampled from each cluster's position weight matrix (PWM) (S12 Fig and Materials and methods). (C) The total statistical entropy (summed entropies across all 7 amino acid positions) versus novelty (the fraction not found in Round 1) of each set of variants is shown. (D) Amino acid frequencies relative to uniform (1/20 chance of each) for the indicated libraries' LY6A-Fc (left) and LY6C1-Fc binders (right). (E) The UMAP projection of sequence exploration for LY6A- (top row) and LY6C1-binders (bottom row) are mapped onto the same UMAP projection as Figs 1–3; the Round 1 UMAP is reproduced in gray in each plot. Sequences with a $\log_2$

enrichment for production fitness > -1.0 (blue) and both fitness > -1.0 and in vivo $\log_2$ enrichment of > 3 (red) are shown for the Round 2 library (left), saturation mutagenesis (center), and SVAE (right). (F) Each point represents a cluster from (E), using the same cluster boundaries as in Fig 1F and 1G, plotted by cluster size versus the cluster's maximum $\log_2$ enrichment in the binding or transduction assay. $\text{Log}_2$ enrichments were calibrated using control sequences (S11 Fig, S12–S23 Data); no calibration adjustment exceeded 2.0. The underlying data supporting Fig 4 can be found at https://doi.org/10.5281/zenodo.7689794: round2_codons_merged.csv and at https://doi.org/10.5281/zenodo.7689794: SVAE_SM_library_codons_merged.csv. ML, machine learning; SVAE, supervised variational auto-encoder.

(3.1)) of predicted target binding and predicted production fitness (S10B Fig). After virus library production, approximately 25% of the 4K binding-only set was not detected, compared to <1% of the 9K joint score set (S10C Fig). Within the saturation mutagenesis-generated variants, 18.4% and 7.6% of the LY6A- and LY6C1-binding sets were not detected, respectively (S10C Fig).

Overall, the SVAE-generated variants were more diverse than the saturation mutagenesis-generated variants, which were designed around fixed motifs, as assessed by 7-mer entropy (Fig 4C), amino acid frequency (Fig 4D), and UMAP projection (Fig 4E). Both SVAE and saturation mutagenesis approaches generated top performers in in vitro target binding and in vivo brain transduction compared to the top hits from the Round 1 screen (Figs 4F and S15); however, as anticipated, saturation mutagenesis based on a few select motifs produced variants that performed better on average than SVAE-generated variants that more extensively sampled the sequence space. These results demonstrated the inherent tradeoff between these 2 approaches, which are both viable sequence diversification strategies. The SVAE approach can explore more of the sequence space, but produces less performant variants on average. In contrast, saturation mutagenesis can more comprehensively explore the space around a few high performing motifs to identify more hits possessing those motifs. Ultimately, the pull-down assays produced data from a single round of screening that could train ML models that capture sufficient understanding about the relationship between amino acid sequences and target binding performance.

## Discussion

We present a rapid method for enhancing the tropism of AAV vectors by introducing de novo interactions with proteins expressed on target cells. Our approach generated BBB-crossing capsids by first screening for direct in vitro interactions with specific proteins rather than by selecting immediately for in vivo success. This mechanism-focused strategy identified thousands of capsids that specifically bind to the mouse brain endothelial cell surface proteins, LY6A or LY6C1, and many of these capsids exhibited enhanced CNS tropisms when validated in vivo, both in a pooled library and when tested individually. Importantly, the tropisms observed with LY6A- and LY6C1-binding capsids in different mouse strains matched expectations based on the strain-specific expression of these proteins. These results demonstrate how new virus capsid–receptor interactions can be introduced through the addition of short linear insertions into AAV capsid proteins.

In vivo selections typically recover a sparse subset of sequences with potential enhancements imparted through unknown mechanisms, which can be specific to a particular strain or species and therefore not amenable to translational studies. In contrast, we showed that a single round of protein target-binding screening yielded highly reproducible and quantitative data based on a known mechanism of action. We leveraged these high-quality data to conduct saturation mutagenesis and ML-guided exploration of a more diverse target-binding sequence space to nominate additional, novel candidates for subsequent screening. Many of these novel

candidates were found to exhibit high levels of in vivo CNS transduction in the validation library, again, within only 2 rounds of screening. Saturation mutagenesis and ML-guided approaches both proved useful—with saturation mutagenesis comprehensively exploring the diversity around one or a few defined sequence motifs and SVAEs serving to explore a wider set of sequences—when used prior to the identification of the top functioning motifs. This approach of AAV capsid engineering can lead to more judicious use of animals by, first, generating capsid libraries populated with performant variants from in vitro assays and, second, guiding the downstream development of capsids in species where their mechanism of action is conserved.

The high-quality data obtained via our target-specific selection strategy also opens the capsid engineering process to a wide range of different computational approaches, with much more room for function-specific optimization and parameterization. A number of groups have generated diverse libraries of capsids for testing using ML-based approaches [39,40], including unsupervised VAEs [37,40]. While our SVAE uses a standard one-hot encoding scheme that worked well using the high-quality data from the pull-down assays, others have experimented with the use of other encoding schemes such as physicochemical parameters [41] or learned representations [42]. As the ML field and its role in biological research continues to develop, in vitro screening methods, such as the protein target-binding assay used here, that generate high-quality, quantitative data will become crucial to take advantage of increasingly sophisticated computational approaches.

The method of engineering capsids through protein target-binding assays enables screening against a wide range of cell proteins across different species. Recently, Shay and colleagues have identified Carbonic Anhydrase IV [43] as the cellular protein co-opted by the 2 previously reported mouse BBB-crossing AAVs (9P31 and 9P36) that we reported here do not engage LY6A or LY6C1 [3]. Like LY6A and LY6C1, Carbonic Anhydrase IV is a GPI-anchored protein that is highly expressed on CNS endothelial cells. Notably, this protein is present in both rodents and primates, and may therefore represent a prime target for engineering receptor-targeted AAVs with a predictable mechanism of action for human CNS gene therapy. Traits necessary for a protein to function as a new AAV receptor may include high levels of cell surface exposure, specific intracellular trafficking routes, and a propensity for transcytosis in the case of interactions that mediate the crossing of vascular barriers. However, these traits alone may not predict whether a protein is capable of facilitating AAV entry into and transduction of the cell, and our understanding of the traits that are critical for an AAV receptor will likely expand as more AAV–receptor interactions are described. Encouragingly, the identification of suitable protein targets is aided by recent advances in cell type characterization using single-cell transcriptomics and proteomics, e.g., mouse [44] and human brain vascular atlases [45–47]. As more of the underlying biology of transcytosis and CNS transduction are characterized, mechanism-focused strategies offer a promising avenue for accelerating the development of capsids that bind defined CNS targets conserved across animal models and humans. The in vitro pull-down assay approach demonstrated in this work should enable accelerated capsid library screening across protein targets from multiple species to improve translation, as well as screening across different protein targets to identify capsids with highly specific target binding.

## Materials and methods

### Capsid library cloning

The RNA expression system for the selection of functional AAV capsids was used as previously described [4] with a modification to include a Woodchuck Hepatitis Virus (WHV)

Posttranscriptional Regulatory Element (WPRE) between the restriction enzyme site SalI and HindIII. The wild-type AAV9 capsid gene sequence was synthesized (GenScript) with nucleotide changes at S448 (TCA to TCT, silent mutation), K449R (AAG to AGA), and G594 (GGC to GGT, silent mutation) to introduce XbaI and AgeI restriction enzyme recognition sites for library fragment cloning.

For generating 7-mer NNK libraries, the hand-mixed primer Assembly-NNK-AAV9-588 (CCCGGAAGTATTCCTTGGTTTTGAACCCAACCGGTCTGCGCCTGTGCMNNMNN MNNMNNMNNMNNMNNTTGGGCACTCTGGTGGTT) encoding a 7-mer insertion between amino acid residues 588 and 589 of AAV9 was used as the reverse primer along with the Assembly-XbaI-F oligo (CACTCATCGACCAATACTTGTACTATCTCT) as a forward primer in a PCR reaction using Q5 High-Fidelity 2X Master Mix (NEB #M0492S) following the manufacturer's protocol for 30 cycles with 10 ng pUC57-wtAAV9-X/A plasmid.

To assemble an oligonucleotide Library Synthesis (OLS) Pool (oligo pool; Agilent) into an AAV genome, 5 pM of the OLS pool was used as an initial reverse primer along with 0.5 μm Assembly-XbaI-F oligo as the forward primer to amplify and extend 10 ng pUC57-wtAAV9-X/A for 5 cycles. Then, the reaction was spiked with 0.5 μm of primer Assembly_AgeI-R (GTATTCCTTGGTTTTGAACCCAACCG) and amplified for an additional 25 cycles. The PCR product was purified using a Zymoclean DNA Gel Recovery kit (Zymo Research #D4007) following the manufacturer's protocol. The 7-mer NNK or oligo pool PCR products were assembled into the RNA expression plasmid as previously described [2].

## SaCas9 and gRNA plasmids

The SaCas9 vector was derived from AAV-CMV::NLS-Sa-Cas9-NLS-3xHA-bGHpA;U6::BsaI-sgRNA obtained from Dr. Feng Zhang through Addgene (#61591). The gRNA scaffold sequence was modified as described in [48]. The 2 tandem U6-sgRNA cassettes were added to a GfABC1D-NLS-GFP reporter vector using gBlocks (IDT). The GFAP promoter (GfABC1D) was obtained from M. Brenner and previously described [49].

## Virus production and titering

For both library and individual recombinant AAVs, viruses were generated by triple transfection of HEK293T/17 cells (ATCC, CRL-11268) using polyethylenimine (PEI), purified by ultracentrifugation over iodixanol gradients, and titered as previously described [2,4].

## Fc-fusion cloning and protein purification

The open reading frames of LY6A (NM_001271416.1) and LY6C1 (NM_010741.3) were separately cloned into an expression vector backbone with a C-terminal Fc-tag (Addgene plasmid #115773) using XbaI/EcoRV. The Fc construct DNA was transfected into HEK293T/17 cells (40 μg per 150 mm dish with PEI) in complete DMEM media with 5% FBS, and 12 to 16 h post-transfection, the plate was rinsed with PBS, and serum free media (Lonza, BEBP12-764Q) was added. Media containing the secreted Fc-fusion proteins was collected at 48 and 96 h after the media change, filtered (Millipore SE1M003M00), and stored at 4°C until use. Approximately 35 μL of protein A-conjugated beads (Thermo Fisher, 10001D) and Tween-20 (0.05% final concentration) were added to 30 mL of media and incubated at 4°C with end-to-end rotation. The next day, the beads were washed 3 times with DPBS containing 0.05% Tween-20. Expression was assessed by running a 5 μL aliquot of protein bound beads on a 4% to 12% protein gel; the remaining fraction was used for pull-down assay.

### Pull-down assay

A total of 10 μL of Fc-fusion protein bound beads were mixed with 1e10 vg AAV capsid library in DPBS with 0.05% Tween-20 and 1% BSA and incubated overnight at 4°C. The next day, beads bound with virus were washed 3 times with PBS with 0.05% Tween-20, and then treated with proteinase K to extract the viral genome that was purified with AMPure XP beads following the manufacturer's protocol for PCR recovery and NGS sample preparation.

### In vitro bound vector genome and transduction assays

The cDNA of *Ly6a*, *Ly6c1*, *or* eGFP was transfected into HEK293T cells using PEI in 6-well plates (2 million cells/well) for binding or in 100 mm dishes (10 million cells/plate) for transduction and used for assays 48 h later. For assessments of AAV binding to the transfected HEK293T cells, the cells were chilled to 4°C and the media was exchanged with fresh cold media containing the AAV capsid library at 10,000 vg/cell. One hour later, cells were washed 3 times with cold PBS and then lysed for total DNA extraction and PCR amplification of bound capsid sequences. For the transduction assays, the AAV capsid library was added at 1,000 vg/cell and cells were harvested for RNA extraction after 60 h.

### Animals

All procedures were performed as approved by the Broad Institute or the Baylor College of Medicine (BCM) Institutional Animal Care and Use Committee (IACUC). Ai9 (007909), BALBc (000651) and C57BL/6J mice (000664) were purchased from the Jackson Laboratory (JAX). Intravenous administration of rAAV vectors was performed by injecting the virus into the retro-orbital sinus. Mice treated at the Broad Institute were euthanized with a lethal dose of Euthasol (Virbac) or by cervical dislocation under the influence of isoflurane as approved under our IACUC protocol. Mice at the BCM-Rice SATC were euthanized by cervical dislocation under the influence of isoflurane anesthesia, according to the approved BCM IACUC protocol.

### In vivo screening

For the selection in mice, 1e11 vg of the capsid libraries were intravenously injected into adult female animals. Two weeks post-injection, mice were euthanized and the brain and liver were collected. CNS transduction in vivo was measured by the transcribed capsid mRNAs as previously described [3,4,9]. Briefly, RNA was extracted from tissues with Trizol reagent followed by cleanup with RNeasy kit (Qiagen). Approximately 5 μg of RNA was converted to cDNA using Maxima H Minus Reverse Transcriptase (Thermo Fisher, EP0751) according to manufacturer instructions and the resulting cDNA was used for capsid sequence recovery. In the Round 2 screen, 1 C57BL/6J mouse was excluded due to poor transduction by the virus library.

### Biodistribution and in vivo transduction of SVAE library

Eight-week-old C57BL/6J were injected intravenously with 1e11 vg of the SVAE virus library. For biodistribution, mice were anesthetized, perfused with PBS, and the brain and liver were collected 2 h post-injection. Total DNA including the viral genome was extracted with a DNeasy kit (Qiagen) and used for NGS sample preparation. For in vivo transduction, tissues were harvested 3 weeks post-injection for RNA extraction and capsid sequence recovery with PCR.

## Tissue processing and imaging

Tissues were processed as previously described [11]. Briefly, mice were anesthetized and transcardially perfused first with phosphate buffered saline (PBS, pH 7.4) at room temperature and then with freshly prepared 4% paraformaldehyde (PFA) in PBS. The tissues were fixed in 4% PFA overnight and afterwards transferred to PBS with 0.05% sodium azide. Sagittal brain sections (50 to 100 μm) were prepared with a vibratome (Leica). IHC was performed with either S100 antibody (Abcam, Ab52642) diluted 1:250, NeuN antibody (Invitrogen, MA5-33103) or APC antibody (OP80, Millipore Sigma) in PBS containing 5% donkey serum, 0.1% Triton X-100, 0.05% sodium azide incubated overnight at room temperature on a laboratory rocker. Sections were then washed 3 times with PBS containing 0.1% Triton X-100 for 5 min. A secondary antibody (Invitrogen, A32728) or (Invitrogen, A32795) was used at 1:500 and incubated and washed in similar conditions to the primary antibody. In the SaCas9 experiment, the quantification of the fraction of the area occupied by functionally edited astrocytes (tdTomato-positive) was performed by thresholding in Fiji (Image J). Briefly, whole-brain sagittal images of tdTomato expression were opened in Fiji, converted to 8-bit, and a background subtraction was performed (rolling ball radius 50.0). The images were then subjected to a manual threshold setting that was set to ensure that astrocyte soma and processes, but not background non-transduced areas reached the threshold. Once set, the threshold was kept constant across all images. The editing validation study (BCM-Rice) was performed on brain tissue samples that were fixed in 4% PFA overnight, equilibrated in 30% sucrose, frozen in optimal cutting temperature (OCT) compound, and sectioned sagittally. Native fluorescence images were obtained on an AxioScan.X1 slide scanner. For additional detail and examples, see Deverman_method_for_area_based_quantification_of_editing_efficiency.pdf (https://scge.mcw.edu/toolkit/download/1036).

## NGS sample preparation

To prepare AAV libraries for sequencing, qPCR was performed on extracted AAV genomes or transcripts to determine the cycle thresholds for each sample type to prevent overamplification. Once cycle thresholds were determined, a first round PCR amplification using equal primer pairs (1–8) (Table 1) were used to attach Illumina Read 1 and Read 2 sequences using

**Table 1. PCR1 primers.**

| Name | 5' Handle | Sequence |
|---|---|---|
| seq1_F | Read 1 | CTTTCCCTACACGACGCTCTTCCGATCTNNNNNNNNCCAACGAAGAAGAAATTAAAACTACTAACCCG |
| seq2_F | Read 1 | CTTTCCCTACACGACGCTCTTCCGATCTNNNNNNNCCAACGAAGAAGAAATTAAAACTACTAACCCG |
| seq3_F | Read 1 | CTTTCCCTACACGACGCTCTTCCGATCTNNNNNNCCAACGAAGAAGAAATTAAAACTACTAACCCG |
| seq4_F | Read 1 | CTTTCCCTACACGACGCTCTTCCGATCTNNNNNCCAACGAAGAAGAAATTAAAACTACTAACCCG |
| seq5_F | Read 1 | CTTTCCCTACACGACGCTCTTCCGATCTNNNNCCAACGAAGAAGAAATTAAAACTACTAACCCG |
| seq6_F | Read 1 | CTTTCCCTACACGACGCTCTTCCGATCTNNNCCAACGAAGAAGAAATTAAAACTACTAACCCG |
| seq7_F | Read 1 | CTTTCCCTACACGACGCTCTTCCGATCTNNCCAACGAAGAAGAAATTAAAACTACTAACCCG |
| seq8_F | Read 1 | CTTTCCCTACACGACGCTCTTCCGATCTNCCAACGAAGAAGAAATTAAAACTACTAACCCG |
| seq1_R | Read 2 | GGAGTTCAGACGTGTGCTCTTCCGATCTCATCTCTGTCCTGCCAAACCATACC |
| seq2_R | Read 2 | GGAGTTCAGACGTGTGCTCTTCCGATCTNCATCTCTGTCCTGCCAAACCATACC |
| seq3_R | Read 2 | GGAGTTCAGACGTGTGCTCTTCCGATCTNNCATCTCTGTCCTGCCAAACCATACC |
| seq4_R | Read 2 | GGAGTTCAGACGTGTGCTCTTCCGATCTNNNCATCTCTGTCCTGCCAAACCATACC |
| seq5_R | Read 2 | GGAGTTCAGACGTGTGCTCTTCCGATCTNNNNCATCTCTGTCCTGCCAAACCATACC |
| seq6_R | Read 2 | GGAGTTCAGACGTGTGCTCTTCCGATCTNNNNNCATCTCTGTCCTGCCAAACCATACC |
| seq7_R | Read 2 | GGAGTTCAGACGTGTGCTCTTCCGATCTNNNNNNCATCTCTGTCCTGCCAAACCATACC |
| seq8_R | Read 2 | GGAGTTCAGACGTGTGCTCTTCCGATCTNNNNNNNCATCTCTGTCCTGCCAAACCATACC |

Q5 Hot Start High-Fidelity 2X Master Mix with an annealing temperature of 65˚C for 20 s and an extension time of 1 min. Round 1 PCR products were purified using AMPure XP beads following the manufacturer's protocol and eluted in 25 μL UltraPure Water (Thermo Scientific); then, 2 μL was used as input in a second round PCR amplification to attach Illumina adaptors and dual index primers (NEB, E7600S) for 5 PCR cycles using Q5 HotStart-High-Fidelity 2X Master Mix with an annealing temperature of 65˚C for 20 s and an extension time of 1 min. The second round PCR products were purified using AMPure XP beads following the manufacturer's protocol and eluted in 25 μL UltraPure DNase/RNase-Free distilled water (Thermo Scientific).

To quantify the amount of second round PCR product for NGS, an Agilent High Sensitivity DNA Kit (Agilent, 5067–4626) was used with an Agilent 2100 Bioanalyzer system. PCR products were then pooled and diluted to 2 to 4 nM in 10 mM Tris-HCl (pH 8.5) and sequenced on an Illumina NextSeq 550 following the manufacturer's instructions using a NextSeq 500/550 Mid or High Output Kit (Illumina, 20024904 or 20024907), or on an Illumina NextSeq 1000 following the manufacturer's instructions using NextSeq P2 v3 kits (Illumina, 20046812). Reads were allocated as follows: I1: 8, I2: 8, R1: 150, R2: 0.

## NGS data processing

Sequencing data was demultiplexed with *bcl2fastq* (version v2.20.0.422) using the default parameters. The Read 1 sequence (excluding Illumina barcodes) was aligned to a short reference sequence of AAV9:

CCAACGAAGAAGAAATTAAAACTACTAACCCGGTAGCAACGGAGTCCTATGGA
CAAGTGGCCACAAACCACCAGAGTGCCCAA<u>NNNNNNNNNNNNNNNNNNNNN</u>G
CACAGGCGCAGACCGGTTGGGTTCAAAACCAAGGAATACTTCCG

Alignment was performed with *bowtie2* (version 2.4.1) [50] with the following parameters:
--end-to-end --very-sensitive --np 0 --n-ceil L,21,0.5 --xeq -N 1 --reorder --score-min L,-0.6,-0.6–5 8–3 8

The resulting sam files from *bowtie2* were sorted by read and compressed to bam files with *samtools* (version 1.11-2-g26d7c73, *htslib* version 1.11-9-g2264113) [51,52].

*Python* (version 3.8.3) scripts and *pysam* (version 0.15.4) were used to flexibly extract the 21-nucleotide insertion from each amplicon read. Each read was assigned to one of the following bins: Failed, Invalid, or Valid. Failed reads were defined as reads that did not align to the reference sequence or that had an in/del in the insertion region (i.e., 20 bases instead of 21 bases). Invalid reads were defined as reads whose 21 bases were successfully extracted, but matched any of the following conditions: (1) Any one base of the 21 bases had a quality score (AKA Phred score, QScore) below 20, i.e., error probability > 1/100; (2) any one base was undetermined, i.e., "N"; (3) the 21 base sequence was not from the synthetic library (this case does not apply to NNK library); or (4) the 21 base sequence did not match a pattern, i.e., NNK (this case does not apply to the synthetic libraries). Valid reads were defined as reads that did not fit into either the Failed or Invalid bins. The Failed and Invalid reads were collected and analyzed for quality control purposes, and all subsequent analyses were performed on the Valid reads.

Count data for valid reads was aggregated per sequence, per sample, and was stored in a pivot table format, with nucleotide sequences on the rows, and samples (Illumina barcodes) on the columns. Sequences not detected in samples were assigned a count of 0.

To minimize the effect of sequencing error on analysis of the library data, variants in Library 1 and Library 2 with fewer than 10 total read counts across all samples and assays (including those not described in this paper) were excluded.

## Data normalization

Count data was reads per million (RPM) normalized to the sequencing depth of each sample $j$ (Illumina barcode) with:

$$r_{i,j} = \frac{k_{i,j}}{\sum_{l=1}^{n} k_{l,j}} \times 1000000. \tag{1.1}$$

Where $r$ is the RPM-normalized count, $k$ is the raw count, $i = 1, \ldots, n$ sequences, and $j = 1, \ldots, m$ samples.

As each biological sample was run in triplicate, we aggregated data for each sample by taking the mean of the RPMs:

$$\mu_{i,s} = \frac{\sum_{l=1}^{p} r_{i,l}}{p} \tag{1.2}$$

across $p$ replicates of sample $s$. We estimated normalized variance across replicates by taking the coefficient of variation (CV):

$$CV_{i,s} = \frac{\mu_{i,s}}{\sigma_{i,s}}, \tag{1.3}$$

where $\sigma_{i,s}$ is the standard deviation for variant $i$ in sample $s$ over $p$ replicates.

Log$_2$ enrichment for each sequence was defined as:

$$e_{i,s} = \log_2\left(\frac{\mu_{i,s} + \lambda}{\mu_{i,t} + \lambda}\right), \tag{1.4}$$

where $\lambda$ is a pseudocount constant for ensuring valid values for the log transformation. For all data analyses, $\lambda$ is set to 0.01.

## Clustering analysis

Target-specific capsids for LY6A and LY6C1 were selected according to their log$_2$ enrichment for their respective receptor and the Fc-only control (blue highlighted regions in Figs 1D and S1A). The log$_2$ enrichment cutoffs used for this analysis were: target $> 8$ when Fc enrichment missing (left marginal plot), target $> 8$ for Fc enrichment $< = 0$, target $> (9/17 \, {}^{*}$ Fc enrichment) $+ 8$ for Fc enrichment $> 0$. This inclusion threshold yielded $n = 5,724$ and $n = 2,291$ LY6A-specific capsids for Library 1 and Library 2, respectively, and $n = 7,170$ and $n = 4,214$ LY6C1-specific capsids for Library 1 and Library 2, respectively. LY6A- and LY6C1-specific sequences were then separated from each other, and capsids from each target set were analyzed separately. Capsid sequences were one-hot encoded into vectors of length $20 \times 7 = 140$, and projected with UMAP with the following parameters: n_components = 2, n_neighbors = 200, min_dist = 0.15, metric = Euclidean. Capsid sequences were then clustered separately for LY6A and LY6C1, using their UMAP projection values (X1, X2) with the GaussianMixture model from scikit-learn [53] with parameters n_components = 40, random_state = 1, n_init = 10, max_iter = 1,000.

## Inter-library calibration

We produced and sequenced the Round 2 and combination SVAE/saturation mutagenesis libraries separately. The enrichment score of a variant in a library is derived from comparisons with the other members of the library, meaning that enrichment is a relative value. Thus,

enrichments are comparable *within* a library, but not directly to other libraries. To enable comparisons between our 2 libraries, we included 3,352 variants in both for use in calibration of enrichment scores. We use a simple calibration method which adjusts enrichment scores to minimize the sum of errors between all shared variants.

S11 Fig shows the shared variants' enrichments (A–C) and enrichment distribution after calibration (D). In addition to different library members, there is variation in the sequencing depth of the 2 libraries. Our calibration method does not account for sequencing depth, which we hypothesize causes some poorly enriched variants to show large enrichment differences between libraries (C, green box), so we chose to drop those variants from the calibration. Variants only detected in one library were also excluded from calibration. Note that while calibration improves comparisons between the libraries, error remains and can be substantial with standard deviations of 2.7, 2.1, and 2.3 for LY6A binding, LY6C1 binding, and the brain transduction assay respectively.

## Synthetic oligo pool library design and synthesis

The synthetic oligo pool library used for the secondary screening assay (Round 2) was obtained from Agilent. The oligonucleotides were designed to conform to the same template binding and assembly overlapping sequences as described above for the Round 1 NNK primers. The library oligo pool consisted of 7-mer insertion sequences recovered from the Round 1 pull-down assays based on the following criteria: (Library 1) Target $\log_2$ enr > 5, Target-Fc $\log_2$ enr—Fc control $\log_2$ enr > 3, and Target-Fc $\log_2$ enr—bead only control $\log_2$ enr >3 and were detected on at least 2 of the 3 replicates; (Library 2) Target $\log_2$ enr > 6 and filtration for specificity versus all other controls and assays based on Target RPM/SUM of all counts. The library also contained all of the top sequences recovered from the Round 1 C57BL/6J and BALB/cJ transcribed capsid sequence screen, published reference sequences and additional sequences screened for LY6A and LY6C1 binding through additional studies not described in this study. All sequences were encoded by 2 distinct nucleotide sequences designed to serve as biological replicates.

## Individual capsid characterization

Individual capsids were cloned into an iCAP-AAV9 (K449R) backbone (GenScript), produced as described above with a DNA genome that encodes nuclear localized GFP under a CAG promoter, and administered to C57BL/6J or BALB/cJ (Jackson Laboratory, 000664) mice at a dose of $3 \times 10^{11}$ vg/mouse. Three weeks later, mice were perfused with 4% PFA. Tissue processing, immunohistochemistry, and imaging were performed as previously described [11]. Recombinant AAV-BI28 vectors were administered to adult male and female mice via the retro-orbital sinus. No mice were excluded from the analyses. Experimenters were not blinded to sample groups.

## SVAE model

The data used in training the SVAE were of the form

$$d_i = (x_i, y_i, cv_i),$$

where $x_i$ is a one-hot encoded 7-mer AA sequence, $y_i$ the corresponding log2enr value (Eq 4) in the target assay, and $cv_i$ the coefficient of variation (Eq 3). Only data points with assay mean RPM > 0 were included in training (at least 1/3 replicates had to be detected). The training/validation split was 0.8 and 0.2, respectively.

**SVAE model architecture.** The SVAE (S12 and S13 Figs) is composed of the following 3 neural network modules:

$$X \to Z \text{ encoder}$$

$$Z \to X \text{ decoder}$$

$$(X, Z) \to Y \text{ regressor.}$$

Among these, the encoder and decoder together form a standard VAE; the addition of the regressor enables supervision. The encoder learns a map

$$enc : X \to Z,$$

where Z is a latent space subject to the standard Gaussian prior [54]. The decoder learns a map

$$dec : Z \to X$$

from the latent space back to the original (one-hot-encoded) sequence space. The regressor learns a map

$$reg : (X, Z) \to Y = \mathbb{R},$$

which takes as input a combined representation of a sequence $x_i$ and its (learned) latent representation $z_i$, and maps it to a predicted $\log_2$ enrichment value $\hat{y}_i$.

In our model,

$$X = [0, 1]^{7 \times 20} = [0, 1]^{140}$$

(such that 7-mer AA sequences are one-hot-encoded with respect to an alphabet of 20 amino acids),

$$Z = \mathbb{R}^2$$

(i.e., we used a 2-dimensional latent space), and

$$Y = \mathbb{R}.$$

For our encoder, we used a 2-hidden-layer fully connected neural network with 100 and 40 nodes in the hidden layers, respectively, with ELU activation. Our decoder, constructed in mirror image of the encoder, was a 2-hidden-layer fully connected neural network with 40 and 100 nodes in the hidden layers, respectively. Our regressor was again a 2-hidden-layer fully connected neural network, but with 100 and 10 nodes in the hidden layers, respectively.

**VAE training.** The encoder and decoder networks are trained jointly with respect to the reconstruction loss

$$\mathcal{L}_{recon}(x_i, y_i) = \text{CEL}(x_i, dec(enc(x_i))), \tag{2.1}$$

where CEL is the standard cross-entropy loss.

The regressor is trained with respect to the regression loss

$$\mathcal{L}_{regr}(x_i, y_i, cv_i) = |y_i - reg(x_i, enc(x_i))|/cv_i. \tag{2.2}$$

Additionally, there is a distributional loss term: $L_{dist}$, computed as the KL divergence of the VAE latent space and a standard Gaussian prior [54].

$$L_{dist} = \text{KL}(\text{VAE latent space, standard gaussian prior}). \tag{2.3}$$

The overall loss of the SVAE is a linear combination of the (1) reconstruction loss; (2) regression loss; and (3) a distributional loss.

$$L_{overall} = \alpha(L_{recon}) + \beta(L_{regr}) + \gamma(L_{dist}) \tag{2.4}$$

Where $\alpha = 1.0$, $\beta = 0.5$, and $\gamma = 0.1$ parameterize the extent to which each loss term factors into the overall loss. These were tuned with hyperparameter optimization with the goal of producing a coherent latent space that separates the regressor values along a gradient.

Both models were trained until convergence, with a convergence threshold of 0.005. Convergence was identified as when the maximum difference between consecutive epochs across all loss metrics ($L_{overall}$, $L_{recon}$, $L_{regr}$, and $L_{dist}$) is less than the convergence threshold for 3 of 5 consecutive epochs. When trained according to this convergence criterion The LY6A-Fc and LY6C1-Fc model training ran for 64 and 63 epochs, respectively.

## SVAE sequence generation

After training, each model's training data was projected into its trained 2D latent space. These points were clustered into 5 primary clusters using KMeans, using both latent space coordinates and $\log_2$ enrichment (S14 Fig). The incorporation of the regression loss into training encourages points to separate spatially by enrichment value, along a gradient.

For each cluster, we calculated the mean enrichment of the sequences contained within it; because the latent space was encouraged to separate enrichment values along a gradient, and clustering was done using both latent space coordinates and enrichment value, the primary clusters formed clear high-, medium-, and low-mean-enrichment clusters (S14 Fig). We isolated the single cluster with the highest mean enrichment to serve as the basis distribution for generating new sequences. This top cluster was then re-clustered with K-means into 10 subclusters. Because the VAE's latent space is trained to encode both sequence and corresponding assay enrichment, these subclusters correspond approximately to motif regions within the high performing cluster.

To generate new variants, for each subcluster, we encoded the amino acid frequencies at each position in the form of a position weight matrix (PWM). At each position, amino acids whose frequency was below the 80th percentile were filtered out. Using the remaining set of passing AAs per position, we generated all possible combinations of 7-mers, and 7-mers already present in the training data were ignored.

## Optimization library composition

The Optimization Round 2 library consists of 96K capsid variants (two 7-mer AA replicates per variant, 192K DNA sequences total). This library of 96K variants comprises: 64K Saturation Mutagenesis, 26K SVAE-generated, 50 published/internal controls, 1K stop-codon controls, 6K calibration controls, and 4K positive training controls (S10A Fig). Within the same experimental pool as the Optimization Round 2 library were included an additional 26K sequences generated by an alternative VAE-based generation scheme. These additional sequences were used to compare the performance of the SVAE-based generation scheme as described in the text with that of the alternative scheme. We chose to present, in the comparison of library selection strategies (alongside saturation mutagenesis and standard selection), the scheme that generated the higher-performing set of variants on average.

The 26K SVAE-generated variants are equally divided between LY6A and LY6C1. Each receptor is further divided into 2 sets of sizes 4K and 9K, respectively, with different selection criteria: (1) the top 4K variants with the highest predicted binding enrichment according to the respective SVAE, and (2) the top 9K variants scoring the highest on a joint score depending

on both high predicted binding enrichment and high predicted production fitness (S10B Fig). In order to compute the joint score across all novel generated variants per receptor, the set of predicted binding values is linearly scaled to lie in a range of [0,1]. The same is done for the fitness values. The scaled values are then simply added together (with equal weight) to compute the joint score. That is, if gen_variants is the full set of novel variants generated for either receptor, for a variant $v$ in gen_variants, the joint score joint_score($v$) of $v$ is defined as follows:

$$joint\_score(v) = \frac{\text{pred\_binding}(v) - \min_{w \in \text{gen\_variants}} \text{pred\_binding}(w)}{\max_{w \in \text{gen\_variants}} \text{pred\_binding}(w) - \min_{w \in \text{gen\_variants}} \text{pred\_binding}(w)}$$
$$+ \frac{\text{fitness}(v) - \min_{w \in \text{gen\_variants}} \text{fitness}(w)}{\max_{w \in \text{gen\_variants}} \text{fitness}(w) - \min_{w \in \text{gen\_variants}} \text{fitness}(w)} \tag{3.1}$$

This split accounts for the SVAE's lack of production fitness knowledge, i.e., the VAE model may not understand the destabilizing effects of certain AAs in the context of our 7-mer insert, such as cysteine (C) or tryptophan (W). The production fitness predictor is described in a separate article [38].

The Top 4K subset, which was chosen on binding enrichment alone, displayed a markedly reduced observed production fitness, and a significant portion (24.3% for LY6A, 27.3% for LY6C1, S10C Fig) were not observed in the produced library—suggesting that these variants' fitness was below our detection threshold. All subsequent analyses with SVAE-generated sequences use only the Top 9K subset.

The 50 reference sequences (S11 Data) include AAV capsids developed both in our lab and by other groups [1–3,30], and 1K variants with stop codons in the 7-mer insert were included to assess the cross-packaging rate. The 6K variants (3K each LY6A, LY6C1) are calibration controls, used to calibrate binding enrichment scores between this library and the training data (Round 1 Library 1). Each set of 3K variants was chosen to cover the dynamic range of each binding enrichment distribution. Finally, 2K variants were included as positive controls, 1K for each receptor, and sampled from the training data used to train each receptor's respective SVAE model.

## Saturation Mutagenesis library generation

The Saturation Mutagenesis library consists of 8 motifs (4 motifs per receptor), with 8K variants per motif (64K total). Each motif has 4/7 positions fixed, leaving 3/7 flexible ($20^3 = 8,000$ possible combinations). Starting from 236,951 sequences from our Round 1 library 1, we decompose each sequence into n-grams (motifs) of length 1–5. Wildcard positions within each motif are permitted with a maximum of 3 non-edge wildcards (e.g., A***A). The motif's starting index (0-indexed) is appended to the end of the motif, to indicate motif position within the 7-mer, e.g., ABCDEFG → BCD1. Using this method, we build a bipartite graph of sequences on one side and motifs on the other, such that each sequence is linked to many motifs, and vice versa.

With this graph, we calculate several summary statistics for each motif: (1) motif "specific length," the number of non-wildcard characters in the motif, e.g., A**A = 2; (2) the number of sequences linked to each motif; (3) a "motif enrichment," the mean binding enrichments of the motif's linked sequences. The specific length and number of sequences is useful for understanding motif specificity in the context of our Round 1 library 1. The more general the motif, the more its motif enrichment trends towards the population average of binding enrichments. For the Saturation Mutagenesis library, we chose motifs with a specific length small enough to

admit thousands of variants per motif under combinatorial generation, but still with enough specificity to have a significant impact on enrichment. We selected a coherent set of motifs that exhibited high enrichment relative to other motifs of the same specificity.

The chosen motifs were PF4 for LY6A and G*S3 for LY6C1. Given the constraints of our library size, we chose to select sub-motifs within these general motifs to fix for saturation mutagenesis. For LY6A, these were: ***KPFI, ***KPFL, ***RPFI, ***RPFL. For LY6C1, these were: ***GWSA, ***GWSS, ***GYSA, ***GYSS.

## Supporting information

**S1 Fig. Identification of target-specific capsids using an independently generated random 7-mer library (Library 2).** (**A**) The variant $\log_2$ enrichment (average RPM between replicates, normalized to the starting library RPM) plotted between LY6A-Fc or LY6C1-Fc versus the Fc-only control. The capsids detected in both assays are displayed in the upper-right quadrant. Missing variants from either assay are displayed in the marginal quadrants. (**B**) The $\log_2$ enrichment of selected variants highlighted in blue in (**A**) with each replicate's enrichment plotted in separate rows ($n = 3$). ND = not detected. The underlying data supporting S1 Fig can be found at https://doi.org/10.5281/zenodo.7689794: library2_pulldown.csv.
(TIF)

**S2 Fig. Replicability for in vitro binding assays and in vivo CNS transduction screens.** (**A–D**) Screen of Library 1 replicability of the $\log_2$ RPM of the (**A**) starting virus library, (**B**) LY6A-Fc, (**C**) LY6C1-Fc, and (**D**) Fc-only control. (**E–H**) Screen of Library 2 replicability of the (**E**) starting virus library, (**F**) LY6A-Fc, (**G**) LY6C1-Fc, and (**H**) Fc-only control. (**I, J**) Replicability of separate RNA extractions ($n = 2$ extractions per mouse strain) within each mouse strain ($n = 2$ mice) for (**I**) BALB/cJ and (**J**) C57BL/6J. The capsids detected in both replicates are displayed in the upper-right quadrant. The missing variants from either replicate are displayed in the marginal quadrants. The underlying data supporting S2A–S2D Fig can be found at https://doi.org/10.5281/zenodo.7689794: library1.csv; S2E–S2H Fig at https://doi.org/10.5281/zenodo.7689794: library2_pulldown.csv.
(TIF)

**S3 Fig. Joint clustering of in vitro binders identified from the 2 libraries.** (**A**) The joint UMAP embedding of target-specific 7-mer sequences with sequences colored according to experiment. (**B**) The clustering (Gaussian mixture model, k = 40) on the joint embedding. (**C, D**) The number (**C**) and percentage (**D**) of 7-mer sequences by the Round 1 pull-down screen from each library, per cluster (sorted from left to right by the number of sequences per cluster). The underlying data supporting S3 Fig can be found at https://doi.org/10.5281/zenodo.7689794: LY6A_joint_umap_l1_l2.csv and at https://doi.org/10.5281/zenodo.7689794: LY6C1_joint_umap_l1_l2.csv.
(TIF)

**S4 Fig. Cluster analysis of the Round 1 target-specific 7-mer sequences.** (**A**) LY6A or (**B**) LY6C1 cluster sequence logos and the corresponding heatmap of $\log_2$ enrichments for sequences in each cluster for the Fc-only control, LY6A-Fc, and LY6C1-Fc. The underlying data supporting S4 Fig can be found at https://doi.org/10.5281/zenodo.7689794: LY6A_joint_umap_l1_l2.csv and at https://doi.org/10.5281/zenodo.7689794: LY6C1_joint_umap_l1_l2.csv.
(TIF)

**S5 Fig. The Pearson correlations of Round 2 in vitro and in vivo replicates.** The plots show the replicability of the $\log_2$ reads per million (RPM) of the (**A**) DNA (plasmid) library, (**B**) virus library, (**C**) LY6A-Fc, (**D**) LY6C1-Fc, and (**E**) Fc-only control. The capsids detected in both replicates are displayed in the upper-right quadrant. The missing variants from either replicate are displayed in the marginal quadrants. The replicability of separate RNA extractions are shown for (**F**) BALB/cJ (4 mice [F1, F2, M1, M2], $n = 3$ extraction replicates per animal) and (**G**) C57BL/6J (3 mice [F1, M1, M2], $n = 3$ extraction replicates per animal). The mean RPM from the extraction replicates between animals were compared for (**H**) BALB/cJ and (**I**) C57BL/6J. The underlying data supporting S5 Fig can be found at https://doi.org/10.5281/ zenodo.7689794: round2_codons_merged.csv.
(TIF)

**S6 Fig. The Pearson correlations between AA replicates in the Round 2 library.** The values shown are $\log_2$ RPM for the DNA library and virus library samples, and $\log_2$ enrichment for the in vitro and in vivo samples. Sequences within pairs of 7-mer AA replicates (codon 1 and codon 2) were randomly assigned to either the x- or the y-axis, with the exception of AA sequences missing their partner within a replicate pair that are assigned to the x-axis and plotted in the histogram below each plot. The underlying data supporting S6 Fig can be found at https://doi.org/10.5281/zenodo.7689794: round2_codons_separate.csv.
(TIF)

**S7 Fig. Identification of a brain-enriched motif that binds to neither LY6A-Fc nor LY6C1-Fc.** (**A**) Round 2 variants identified in the Round 1 in vivo screen (red) were filtered by the thresholds shown for low binding to LY6A-Fc (LY6A-Fc binders are shown in gray), low binding to LY6C1-Fc (LY6C1-Fc binders are shown in gray), and high CNS transduction in either C57BL/6J or BALB/cJ mice. This combined filtering yielded 180 variants. (**B**) Hierarchical clustering of the 180 variants by hamming distance (linkage = average, cutoff = 5) yielded 1 large cluster (red, center, $n = 39$). (**C**) $\log_2$ enrichment is shown for each variant ordered by the clustering tree in (**B**) for in vitro binding of the Fc-only control, LY6A-Fc, LY6C1-Fc, and CNS transduction in BALB/cJ or C57BL/6J mice. (**D**) The sequence motif of the center red cluster ($n = 39$) in (**B**) shows a clear pattern of *N*[T/V/I][R/K]**. The underlying data supporting S7 Fig can be found at https://doi.org/10.5281/zenodo.7689794: round2_codons_ merged.csv.
(TIF)

**S8 Fig. Additional characterization of the in vitro screened LY6A and LY6C1-binding capsids.** Binding (**A**) and transduction (**B**) of HEK293 cells expressing *Ly6a*, *Ly6c1*, or a control (GFP) cDNA by the indicated AAVs. (**A, B**) The enrichment of the reference capsids (AAV9, AAV-PHP.eB, or AAVF) in comparison with the capsids identified in this study (BI48, BI49, BI28, BI62, BI65) observed in a pooled library study are shown. (**C**) Liver transduction by the indicated variants from the same library tested in BALB/cJ and C57BL/6J mice is shown. Each graph shows the mean enrichment (bars) of each capsid normalized to AAV9, with individual values from 7-mer AA replicates (encoded by different nucleotide sequences) shown as individual data points (circles) ($n = 4$ animals/per group). The underlying data supporting S8A Fig can be found at https://doi.org/10.5281/zenodo.7689794: S8_A_HEK_binding.csv; S8B Fig at https://doi.org/10.5281/zenodo.7689794: S8_B_HEK_transduction.csv; S8C Fig at https://doi. org/10.5281/zenodo.7689794: S8_C_liver_transduction.csv.
(TIF)

**S9 Fig. BI28 mediates transduction of neurons and glia and in vivo editing in brain astrocytes.** (**A–C**) Immunofluorescence analysis of transduced cell types 4 weeks after adult

intravenous administration of $1 \times 10^{11}$ vg/animal AAV-BI28:CAG-NLS-mScarlet-2A-Luciferase-WPRE-pA. Representative images show colocalization of mScarlet+ cells (magenta) with NeuN (**A**, green) and S100 (**B**, green) in the cerebral cortex and CC1+ cells (**C**, green) in the corpus callosum. Arrows highlight example transduced marker+ cells. Scale bar is 50 μm. (**D**) The schematic shows the dual AAV gene editing system designed to remove the stop cassette and turn on tdTomato expression in Ai9 reporter knock-in mice. The first rAAV expresses SaCas9 from the astrocyte-selective GfABC$_1$D promoter. The second rAAV expresses GFP from the same promoter and 2 tandem U6-driven gRNAs (L1 and R2, with the indicated spacer and PAM sequences). Both rAAV genomes were packaged into AAV-BI28 and co-administered to Ai9 tdTomato reporter mice at $3 \times 10^{11}$ vg/mouse (total dose $6 \times 10^{11}$ vg/mouse) and editing was assessed 4 weeks later. (**E**) Representative whole sagittal brain section images (top: Deverman laboratory results, bottom: BCM-Rice SATC external validation) show Ai9 locus editing as assessed by tdTomato native fluorescence. (**F**) Colocalization of tdTomato expression with GFP (transduction marker) and S100 in cortical astrocytes are shown. (**G**) Independent validation of gene editing with the AAV-BI28 vectors in Ai9 mice by researchers at the BCM-Rice SATC in coordination with the NIH Somatic Cell Genome Editing Consortium. (**H**) The graph shows the quantification of CNS astrocyte editing measured by the fraction of tdTomato+ area above threshold within the indicated brain regions (mean ± SD, Internal cohort, $n = 6$ females and $n = 4$ males; Baylor validation cohort, $n = 6$ females and $n = 6$ males). (**I**) Whole sagittal brain sections show that administration of both vectors is required to achieve gene editing. Note, the weak signal present in both red and green channels that is most notable in the cerebellum in the SaCas9 vector alone brain image is tissue auto-fluorescence rather than GFP expression. The underlying data supporting S9 Fig can be found at https://doi.org/10.5281/zenodo.7689794: S9_fig_data.csv.
(TIF)

**S10 Fig. SVAE library composition and selection of SVAE-generated variants.** (**A**) The combined SVAE and saturation mutagenesis library is composed of saturation mutagenesis variants generated from 1 motif per target (LY6A \*\*\*[K/R]PF[I/L], LY6C1 \*\*\*G[W/Y]S[A/S]) with 32K variants per motif; 13K SVAE-generated variants per target; 50 previously characterized variants from our group and the literature; 1K variants with stop codons to assess cross-packaging; 6K variants (3K per target) that were evenly selected across low-to-high enrichment bins to calibrate the enrichment scores from this library to the library used to train the SVAE models (Round 1, Library 1); 2K variants (1K for each target) that were randomly chosen from the SVAE training data (i.e., variants with non-zero RPM from Round 1) as training data controls. (**B**) The predicted binding enrichment and predicted production fitness for SVAE-generated variants (150K generated in silico per target) are shown. Included in the SVAE Library were the 4K variants with the top predicted binding enrichment according to the SVAE (red), as well as the top 9K variants according to a joint score of predicted binding enrichment and predicted production fitness (yellow). (**C**) The virus library shown in (**A**) was produced and the distributions of the measured production fitness of the saturation mutagenesis-generated and SVAE-generated variants are shown. The underlying data supporting S10B Fig can be found at https://doi.org/10.5281/zenodo.7689794: LY6A_SVAE_generated_sequences.csv and 10.5281/zenodo.7689794: LY6C1_SVAE_generated_sequences.csv; S10C Fig at https://doi.org/10.5281/zenodo.7689794: SVAE_SM_library_codons_separate.csv.
(TIF)

**S11 Fig. The calibration of the SVAE and saturation mutagenesis library to the Round 2 library.** Library variant enrichment scores are relative because they are derived from comparisons to the other members of the same library. The scores of variants from separate libraries

were calibrated by computing a single value that adjusts these scatter plots (**A–C**) on the y-axis to minimize error (see Materials and methods, S12–S23 Data). (**A, B**) Plots of the enrichment scores within the SVAE library versus the Round 2 library for the uncalibrated LY6A- and LY6C1-binding sequences that are common to both libraries. Histograms on the top and right margins show the distribution of total variants (blue) and variants missing in one of the assays (red). (**C**) The same as in (**A, B**), but for the brain transduction assay. For the brain transduction assay, both libraries contain LY6A- and LY6C1-binding variants so a single calibration value was applied. Points in the green box were dropped when computing the calibration. We hypothesize that this discrepancy arose from the Round 2 library being sequenced more deeply. (**D**) Histograms of pre- and post-calibration enrichment for each assay. Calibration values are as follows: LY6A: −0.37, LY6C1: 1.50, brain transduction, combined LY6A/LY6C1: 0.14. The amount of shift between the pre- and post-calibration histograms corresponds to the calibration value for each assay. The underlying data supporting S11 Fig can be found at https://doi.org/10.5281/zenodo.7689794: round2_codons_merged.csv and 10.5281/zenodo.7689794: SVAE_SM_library_codons_merged.csv.
(TIF)

**S12 Fig. SVAE-based sequence generation procedure, SVAE model, and latent spaces.** (**A**) A schematic of the complete SVAE-based sequence generation procedure, including (1) the processing of training data, (2) SVAE training, and (3) sequence generation using the SVAE latent space.
(TIF)

**S13 Fig. An expanded schematic of SVAE training.**
(TIF)

**S14 Fig. Plots of the LY6A-Fc and LY6C1-Fc training points in latent space.** From left to right: all training points colored by assay $\log_2$ enrichment; all training points colored by mean primary cluster (see Materials and methods, SVAE Sequence Generation) $\log_2$ enrichment; top (highest mean enrichment) cluster colored by assay $\log_2$ enrichment; top cluster further clustered into subclusters, colored by mean subcluster $\log_2$ enrichment. The first 3 plots from the left share spatial axes and color scale; the rightmost subclustering plot is centered on its own axes and recolored on its own scale. The underlying data supporting S14 Fig can be found at https://doi.org/10.5281/zenodo.7689794: LY6A_SVAE_training_predictions.csv and at https://doi.org/10.5281/zenodo.7689794 LY6C1_SVAE_training_predictions.csv.
(TIF)

**S15 Fig. The in vitro binding and brain transduction enrichment scores of individual Round 2, saturation mutagenesis, and SVAE variants.** The data from Fig 4F (S12–S23 Data) is shown without clustering. The panels show the performance of variants for each assay: LY6A-Fc pull-down, LY6C1-Fc pull-down, C57BL/6J mouse brain transduction by LY6A-binding variants, or C57BL/6J mouse brain transduction by LY6C1-binding variants. As in Fig 4F, variants with a fitness value below $\log_2$ enrichment of −1.0 are excluded. The underlying data supporting S15 Fig can be found at https://doi.org/10.5281/zenodo.7689794: round2_codons_merged.csv and at https://doi.org/10.5281/zenodo.7689794: SVAE_SM_library_codons_merged.csv.
(TIF)

**S1 Data. Library 1 UMAP Clusters.**
(CSV)

**S2 Data. Library 1 LY6A UMAP cluster sequences.**
(CSV)

**S3 Data. Library 1 LY6C1 UMAP cluster sequences.**
(CSV)

**S4 Data. Library 2 UMAP clusters.**
(CSV)

**S5 Data. Library 2 LY6A UMAP cluster sequences.**
(CSV)

**S6 Data. Library 2 LY6C1 UMAP cluster sequences.**
(CSV)

**S7 Data. PHP.B-like UMAP clusters and sequences.**
(CSV)

**S8 Data. Round 2 Library reference sequences.**
(CSV)

**S9 Data. In vivo non-LY6A or -LY6C1 binding sequences.**
(CSV)

**S10 Data. Round 2 Library UMAP clusters.**
(CSV)

**S11 Data. SVAE/Saturation Mutagenesis Library reference sequences.**
(CSV)

**S12 Data. Fig 4F Brain transduction LY6A R1 top hits.**
(CSV)

**S13 Data. Fig 4F Brain transduction LY6C1 R1 top hits.**
(CSV)

**S14 Data. Fig 4F Brain transduction LY6C1 saturation mutagenesis table.**
(CSV)

**S15 Data. Fig 4F Brain transduction LY6C1 SVAE table.**
(CSV)

**S16 Data. Fig 4F LY6A R1 top hits.**
(CSV)

**S17 Data. Fig 4F LY6A saturation mutagenesis table.**
(CSV)

**S18 Data. Fig 4F LY6A SVAE table.**
(CSV)

**S19 Data. Fig 4F Brain transduction LY6A saturation mutagenesis table.**
(CSV)

**S20 Data. Fig 4F Brain transduction LY6A SVAE table.**
(CSV)

**S21 Data. Fig 4F LY6C1 R1 top hits.**
(CSV)

**S22 Data. Fig 4F LY6C1 saturation mutagenesis table.**
(CSV)

**S23 Data. Fig 4F LY6C1 SVAE table.**
(CSV)

## Acknowledgments

We thank the members of the Deverman laboratory for continuous discussions of the project; and Alexa E. Martinez, Denise G. Lanza, and John R. Seavitt for providing logistical and technical support for the validation studies at Baylor College of Medicine.

## Author Contributions

**Conceptualization:** Qin Huang, Ken Y. Chan, Fatma-Elzahraa Eid, Benjamin E. Deverman.

**Data curation:** Albert T. Chen, Hikari Sorensen.

**Formal analysis:** Qin Huang, Albert T. Chen, Hikari Sorensen, Andrew J. Barry, Bahar Azari, Fatma-Elzahraa Eid, Christopher J. Walkey, Yujia A. Chan, Benjamin E. Deverman.

**Funding acquisition:** William R. Lagor, Jason D. Heaney, Benjamin E. Deverman.

**Investigation:** Qin Huang, Albert T. Chen, Ken Y. Chan, Hikari Sorensen, Andrew J. Barry, Bahar Azari, Qingxia Zheng, Thomas Beddow, Binhui Zhao, Isabelle G. Tobey, Cynthia Moncada-Reid, Christopher J. Walkey, M. Cecilia Ljungberg, Benjamin E. Deverman.

**Methodology:** Qin Huang, Albert T. Chen, Ken Y. Chan, Hikari Sorensen, Andrew J. Barry, Bahar Azari, Fatma-Elzahraa Eid, Benjamin E. Deverman.

**Supervision:** Benjamin E. Deverman.

**Visualization:** Qin Huang, Albert T. Chen, Hikari Sorensen, Andrew J. Barry, Yujia A. Chan, Benjamin E. Deverman.

**Writing – original draft:** Qin Huang, Albert T. Chen, Hikari Sorensen, Andrew J. Barry, Yujia A. Chan, Benjamin E. Deverman.

**Writing – review & editing:** Qin Huang, Albert T. Chen, Ken Y. Chan, Hikari Sorensen, Andrew J. Barry, Yujia A. Chan, Benjamin E. Deverman.

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
