## [Editor Report · Decision Letter 0]

11 Dec 2022

Dear Dr Deverman, 

Thank you for submitting your manuscript entitled "Targeting AAV vectors to the CNS via de novo engineered capsid-receptor interactions" for consideration as a Research Article by PLOS Biology. Please accept my sincere 

apologies for the delay in getting back to you as we consulted with an academic editor about your submission. 

Your manuscript has now been evaluated by the PLOS Biology editorial staff, as well as by an academic editor with relevant expertise, and I am writing to let you know that we would like to send your submission out for external peer review.

IMPORTANT: After discussing within the editorial team, we feel that your manuscript would be a better fit as a Methods and Resources Article. Upon resubmission (details below), we ask that you please change to this article type. 

Before we can send your manuscript to reviewers, we need you to complete your submission by providing the metadata that is required for full assessment. To this end, please login to Editorial Manager where you will find the paper in the 'Submissions Needing Revisions' folder on your homepage. Please click 'Revise Submission' from the Action Links and complete all additional questions in the submission questionnaire.

Once your full submission is complete, your paper will undergo a series of checks in preparation for peer review. After your manuscript has passed the checks it will be sent out for review. To provide the metadata for your submission, please Login to Editorial Manager (https://www.editorialmanager.com/pbiology) within two working days, i.e. by Dec 13 2022 11:59PM.

Kind regards,

Richard

Richard Hodge, PhD

Associate Editor, PLOS Biology

rhodge@plos.org

PLOS

---

## [Decision Letter · Decision Letter 1]

25 Jan 2023

Dear Dr Deverman,

Thank you for your patience while your manuscript "Targeting AAV vectors to the CNS via de novo engineered capsid-receptor interactions" went through peer-review at PLOS Biology. Please accept my apologies for the delays that you have experienced during the peer review process. Your manuscript has now been evaluated by the PLOS Biology editors, an Academic Editor with relevant expertise, and by three independent reviewers.

The reviews are attached below. As you will see, the reviewers are positive about the AAV capsid generation method and think the manuscript is well done and interesting. In light of these reviews, we are pleased to offer you the opportunity to address the comments from the reviewers in a revision that we anticipate should not take you very long. This includes the pull down validation assays with the Ly6c1 and Ly6a cell lines suggested by Reviewer #2. We will then assess your revised manuscript and your response to the reviewers' comments with our Academic Editor aiming to avoid further rounds of peer-review, although might need to consult with the reviewers, depending on the nature of the revisions.

In addition, we would be grateful if you could address the following data-related and editorial requests that are provided below (A-G):

(A) We would like to suggest that the following modification the title:

"Targeting AAV vectors to the central nervous system by engineering capsid-receptor interactions that enable crossing of the blood-brain barrier” 

(B) In the animal ethics statement in the Methods section, please provide the method of euthanasia used to sacrifice the mice

(C) You may be aware of the PLOS Data Policy, which requires that all data be made available without restriction: http://journals.plos.org/plosbiology/s/data-availability. For more information, please also see this editorial: http://dx.doi.org/10.1371/journal.pbio.1001797

-Supplementary files (e.g., excel). Please ensure that all data files are uploaded as 'Supporting Information' and are invariably referred to (in the manuscript, figure legends, and the Description field when uploading your files) using the following format verbatim: S1 Data, S2 Data, etc. Multiple panels of a single or even several figures can be included as multiple sheets in one excel file that is saved using exactly the following convention: S1_Data.xlsx (using an underscore).

- Deposition in a publicly available repository. Please also provide the accession code or a reviewer link so that we may view your data before publication.

Figure 1B-E, 1H-I, 2B, 2D-F, 3C, 4D, S1A-B, S2A-J, S3C-D, S4A-B, S5A-I, S6, S7A-C, S8C, S9A-D, S12, S13

(D) Please also ensure that each of the relevant figure legends in your manuscript include information on *WHERE THE UNDERLYING DATA CAN BE FOUND*, and ensure your supplemental data file/s has a legend.

(E) Please ensure that your Data Statement in the submission system accurately describes where your data can be found and is in final format, as it will be published as written there.

(F) Please note that per journal policy, we do not allow the mention of "data not shown" or other references to data that is not publicly available or contained within this manuscript (line 399). Please either remove mention of these data or provide figures presenting the results and the data underlying the figure(s).

(G) Please also provide a blurb which (if accepted) will be included in our weekly and monthly Electronic Table of Contents, sent out to readers of PLOS Biology, and may be used to promote your article in social media. The blurb should be about 30-40 words long and is subject to editorial changes. It should, without exaggeration, entice people to read your manuscript. It should not be redundant with the title and should not contain acronyms or abbreviations. For examples, view our author guidelines: https://journals.plos.org/plosbiology/s/revising-your-manuscript#loc-blurb

**IMPORTANT - SUBMITTING YOUR REVISION**

*Resubmission Checklist*

*Published Peer Review*

*PLOS Data Policy*

*Blot and Gel Data Policy*

Sincerely,

Richard

Richard Hodge, PhD

Associate Editor, PLOS Biology

rhodge@plos.org

REVIEWS:

Reviewer #1 (Thomas J McCown, signs review): The manuscript "Targeting AAV vectors to the CNS via de novo engineered capsid-receptor interaction" presents studies that characterize a novel means to engineer novel AAV capsids through in vitro receptor binding. The overall approach arises from a sound rationale and the use of crystallizable fragment domain binding (FC fusions). Given that the receptors for AAV blood brain transit in mice have been identified, these two receptor proteins, LY6A and LY6c were constructed as FC fusions. Then AAV libraries of 7 mer insertions between 588 and 589 of AAV9 VP1 were interacted with the FC fusions and subsequently pull down assays yielded novel capsids that bound either of the two receptor protein. Substantial subsequent studies characterized these novel capsids in comparison to reference capsids such as AAVF and PHPe.B. Extensive in vitro and in vivo studies validated the technique as a means to generate novel AAV capsids that cross the mouse BBB and transduce cells in the mouse brain. As such the findings will be of interest to the AAV gene therapy community. Certainly the findings present a novel means to generate new AAV capsids across species. The only concern involves a method of in vivo validation. In several instances, mRNA determinations were used to indicate in vivo transduction. However, a direct correlation between mRNA and the in vivo amount of viral particles present, as well as the level of protein expression remains tenuous. A tangential example of this arises in figure 3C-D. The enrichment of BI-28 and BI-65 appear virtually identical, but in the individual in vivo validation, GFP expression is vastly different between the two viruses where BI-65 exhibits much lower overall expression compared to BI-28. This divergence should provide an element of caution to the extensive discussion of machine learning applications. Although ML has substantial merit, the final outcome depends upon the content of the learning trials. Also, the general discussion of applicability to other paths of AAV biology should mention that to utilize this technique, it will be important to identify what part of VP1, VP2 and/or VP3 mediate the element to be optimized. In summary, the present studies establish a novel means to generate chimeric AAV capsids that may have enhanced properties and should prove interesting to a wide gene therapy audience. 

Minor -Should define ML when it first appears on line 71

Reviewer #2 (Casey A. Maguire, signs review): The manuscript Targeting AAV vectors to the CNS via de novo engineered capsid-receptor interactions by Huang et al is a proof-of-concept study with the intent of driving selection of efficient AAV vectors for in vivo transduction using in vitro binding assays directed towards specific ligands.

The approach is reminiscent of the early days of phage biopanning to isolate peptides binding to target receptors, which used phage libraries binding to microtiter wells coated with extracellular domains of receptors, except this approach is done using the peptide display library in the context of AAV capsids. The authors purified recombinant Ly6a-Fc and Ly6c-Fc and conjugated these proteins to beads and incubated an AAV9 peptide display library to identify capsids which could bind to either receptor. Using clustering techniques on the NGS data,they then identified motifs that were associated with either Ly6a or Ly6c binding. Finally, they picked a subset of these candidates for in vivo testing in Ly6a expressing mice (C57BL/6) or Ly6c expressing (C57BL/6 and BALB/c) and showed that, as predicted Ly6a binding peptides only transduced C57 and the Ly6c binding peptides transduced both strains of mice, aligning with the receptor expression profile of these mouse strains. 

There are two exciting findings that came about with this work. First, it appears that the Ly6 family of receptors on the BBB of mice is an efficient conduit for AAV-mediated BBB penetration and transduction. This was partially known from the prior studies with PHP.B and Ly6a, but greatly extended by this work to Ly6c and other capsids. The authors appear to unravel the mystery of the BBB-penetration mechanism of many of published engineered capsids from the Gradinaru, Deverman, Maguire, and Nonnenmacher groups, which is very interesting and important to know (particularly in their lack of potential to translate to human for BBB penetration). Second, the use of specific ligands to direct tropism is used successfully here to drive the desired phenotype: BBB penetration and brain transduction after systemic delivery.

Overall, the manuscript is well written, and the extensive experiments and extensive data analysis appear to be well done. This is a very interesting and thought-provoking approach, although it is currently unclear how it will turn out for target ligands that are currently not known to allow the complete transduction process of AAV. However, as more knowledge comes in on the vector biology front, it may be put to good use.

Comments below are suggested to improve the study.

Major Comments.

Comment 1. While the results using Ly6a and Ly6c binding to drive selection of efficient capsids was demonstrated here to work very well at identifying capsids that crossed BBB and transduced the brain, the authors had a priori knowledge that Ly6a would satisfy the entire transduction process. As the authors know (and discuss somewhat), binding to a receptor is only one part of the transduction process and identifying other "AAV-transduction" permissive receptor targets may not be as straightforward. For many years, in vitro selections with AAV vectors have been performed with very little success at translating to desired in vivo transduction properties.

There seems to be a big "what now?" ending to the paper. It would be really useful to understand the authors thinking on how this strategy will be employed in the real world to identify clinically relevant AAV vectors.

 Could the authors speculate on some potential targets or how to enhance this methodology to increase the chances of targeting receptors that would mediate functional transduction? 

Also the statement "This work demonstrates that AAV capsids can be directly targeted to specific proteins to generate potent gene delivery vectors with known mechanisms of action and predictable tropisms" should be tempered somewhat in light of the factor that there was a priori knowledge that the chosen receptor family mediates efficient in vivo transduction in mice. It is currently unknown how this will translate to new receptors that we know exist, just not in the context of AAV transduction permissivity. 

Comment 2. What were the cellular targets in the brain of the identified capsids (BI48, BI49, BI28, BI62, BI65)? Were any "detargeted" from liver?

Comment 3. Could the authors speculate why the Ly6 family of receptors are such promoters of efficient AAV BBB penetration and transduction (at least in mice)?

Comment 4. While the data is highly suggestive of Ly6c pulled-down capsids in vitro mediating transduction using Ly6c receptor in vivo, this has not been definitively demonstrated by the presented data. Could the Ly6c (and for that matter Ly6a) receptor transduction dependence of the identified capsids be demonstrated at least in vitro with the Ly6c1 and Ly6a lines generated in the authors' 2019 PLOS paper?

Minor comments.

Comment 1. It was interesting to see the AAV-F peptide sequence (FVVGQSY) mediates Ly6c binding and many of these amino acids (5/7) were observed in some of the pull-down derived capsids (e.g. FVYGQIA, Fig S6) as well as the MDVIA capsid from the Sabeti group (5/7 amino acid identity). This further suggests that this motif is strongly associated with Ly6C binding and should be included in the text (results or discussion).

Comment 2. Could the authors discuss how this strategy could be used to select for "copy-cat AAV's" in the future and the potential implications? For example if a new receptor is identified by group A to mediate efficient AAV transduction of a particular capsid, group B could use this in vitro pull down technology to identify capsids with similar properties. 

Reviewer #3: Huang et al. provide a detailed quantitative evaluation of AAV capsid variant selection based off of in vitro affinity based binding followed by in vivo validation in two strains of mice known to differentially express the receptor Ly6A. Importantly, the authors evaluate the ability of a SVAE Machine-learning model to generate an AAV capsid variant library enriched for production-competent and high target receptor binding capsids. The approach to select capsids based off of cell-free receptor binding is compelling and has the potential to yield enriched AAV variants that may be rationally pre-selected for transducing in tissues. 

While it is attractive to think that novel AAV capsids with high target cell transduction can be generated with minimal cycling in animal models and at a rapid rate, one potential concern is that binding of specific protein targets may not translate into other steps critical for viral infection such as uptake/cell entry. This aspect should be highlighted as a potential limitation. Another potential limitation is the species-specific difference often seen in such protein based targets (from a sequence perspective) and the implications for species-selectivity that needs to be discussed. Nevertheless, while the study does not provide any significant new insight into AAV biology (from using previously identified receptors in mice), the incorporation of the SVAE machine learning technique exemplifies the power of this technique in being able to predict infectivity and fitness from a vector production standpoint. 

Overall, from a methods/resource contribution perspective, the authors provide a detailed validation of this approach, adequate controls (e.g., comparing and contrasting sequences enriched for Ly6A and Ly6C1 binding compared to Fc binding alone) and the ability of the machine learning approach combined with in-life selection as being on par with in life selection alone. 

Additional comments:

In figure 4F, the recovered enriched sequences are plotted according to cluster log size and cluster max enrichment. Since the data are available and easily sortable, e.g., in Figure S13, the authors should consider plotting enrichment of brain transduction for individual sequences within the SVAE and Sat. mut. Libraries, either in a new Figure 5, or as a subpanel of figure F? This would enable further visual understanding of the performance of the SVAE library in generating individual top enriched amino acid sequences.

Figure S13 as referenced in the text contains the enrichment scores for only the LY6C1 library, while figure 4F clearly plots both the LY6C1 and LY6A libraries. As it stands, there is no textual reference to the quantitative data supporting the LY6A plots in figure 4F. The authors should consider including in the text the reference to the appropriate table listing the LY6A enrichment related to Figure 4F?

---

## [Editor Report · Decision Letter 2]

6 Apr 2023

Dear Dr Deverman,

Thank you for the submission of your revised Methods and Resources Article "Targeting AAV vectors to the central nervous system by engineering capsid-receptor interactions that enable crossing of the blood-brain barrier" for publication in PLOS Biology. On behalf of my colleagues and the Academic Editor, Chaitan Khosla, I am pleased to say that we can accept your manuscript for publication, provided you address any remaining formatting and reporting issues. These will be detailed in an email you should receive within 2-3 business days from our colleagues in the journal operations team; no action is required from you until then. Please note that we will not be able to formally accept your manuscript and schedule it for publication until you have completed any requested changes.

PRESS

Kind regards, 

Richard

Richard Hodge, PhD

Associate Editor, PLOS Biology

rhodge@plos.org

PLOS
